# Leveraging Gradients for Unsupervised Accuracy Estimation under Distribution Shift

**Renchunzi Xie**                                    *renchunzi.xie@ntu.edu.sg*
*College of Computing and Data Science*
*Nanyang Technological University*

**Ambroise Odonnat**                              *ambroise.odonnat@gmail.com*
*Huawei Noah's Ark Lab, Inria$^{\diamond}$*
*Paris, France*

**Vasilii Feofanov**                              *vasilii.feofanov@gmail.com*
*Huawei Noah's Ark Lab*
*Paris, France*

**Ievgen Redko**                                    *ievgen.redko@gmail.com*
*Huawei Noah's Ark Lab*
*Paris, France*

**Jianfeng Zhang**                                  *ievgen.redko@gmail.com*
*Huawei Noah's Ark Lab*
*Shenzhen, China*

**Bo An**[*]                                            *boan@ntu.edu.sg*
*College of Computing and Data Science*
*Nanyang Technological University*

**Reviewed on OpenReview:** *https://openreview.net/forum?id=FIWHRSuoos*

## Abstract

Estimating the test performance of a model, possibly under distribution shift, without having access to the ground-truth labels is a challenging, yet very important problem for the safe deployment of machine learning algorithms in the wild. Existing works mostly rely on information from either the outputs or the extracted features of neural networks to estimate a score that correlates with the ground-truth test accuracy. In this paper, we investigate – both empirically and theoretically – how the information provided by the gradients can be predictive of the ground-truth test accuracy even under distribution shifts. More specifically, we use the norm of classification-layer gradients, backpropagated from the cross-entropy loss after only one gradient step over test data. Our intuition is that these gradients should be of higher magnitude when the model generalizes poorly. We provide the theoretical insights behind our approach and the key ingredients that ensure its empirical success. Extensive experiments conducted with various architectures on diverse distribution shifts demonstrate that our method significantly outperforms current state-of-the-art approaches. The code is available at https://github.com/Renchunzi-Xie/GdScore.

## 1 Introduction

Deploying machine learning models in the real world is often subject to a distribution shift between training and test data. Such a shift may significantly degrade the model's performance at test time (Quinonero-Candela

---
[*]Corresponding Authors.    $^{\diamond}$Univ. Rennes 2, CNRS, IRISA

et al., 2008; Geirhos et al., 2018; Koh et al., 2021) and lead to high risks related to AI satefy (Deng & Zheng, 2021). To alleviate this problem, a common practice is to monitor the model performance regularly by collecting the ground truth of a subset of the current test dataset (Lu et al., 2023). However, this is usually time-consuming and expensive, highlighting the need for unsupervised methods to assess the test performance of models under distribution shift, commonly known as *unsupervised accuracy estimation*.

**Limitations of current approaches.** Current studies mainly focus on outputs or feature representation to derive a test error estimation score. Such a score can represent the calibrated test error or the distribution discrepancy between training and test datasets (Hendrycks & Gimpel, 2016; Guillory et al., 2021; Garg et al., 2022; Deng & Zheng, 2021; Yu et al., 2022b; Lu et al., 2023). For instance, Hendrycks & Gimpel (2016) considered the average maximum softmax score of the test samples as the estimated error. Similarly, Garg et al. (2022) proposed to learn a confidence threshold from the training distribution. Deng & Zheng (2021) quantified the distribution difference between training and test datasets in the feature space, while Yu et al. (2022b) gauges the distribution gap at the parameter level. Although insightful, the current body of literature overlooks a potential tool for test accuracy estimation, namely the gradients that are known to correlate strongly with the generalization error of the deep neural networks (Li et al., 2019; An et al., 2020).

**Why do gradients matter?** Recently, increasing attention has been paid to the intrinsic properties of gradients to design learning algorithms for meta-learning (Finn et al., 2017), domain generalization (Shi et al., 2021; Mansilla et al., 2021) or to improve optimization of DNNs Zhou et al. (2020); Zhao et al. (2022) by relying on them. In domain shift scenarios, Mansilla et al. (2021) proposed to clip the conflicting gradients and introduced a strategy to promote gradient agreement across multiple domains. Meanwhile, Zhao et al. (2022) designed a gradient-based regularization term to make the optimizer find flat minima. In the field of OOD detection, Huang et al. (2021) introduced a gradient-based function to detect out-of-distribution (OOD) samples (relation to our method is compared in Appendix 7). Although the works mentioned above showed the potential of gradients to tackle the learning problems both in OOD and in-distribution (ID) settings, it is still unclear how unsupervised gradient-based scores can correlate with the test accuracy and be used to estimate it. This motivates us to ask:

*Are gradients predictive of test accuracy under distribution shift?*

In this paper, we shed new light on this open question: we surprisingly observe that there exists a strong linear relationship between gradient norm and test accuracy under distribution shift. Moreover, our theoretical analysis shows that the gradient norm conveys information on the generalization capacity of a well-calibrated model. In a nutshell, this work provides direct evidence that gradient-based information correlates with generalization performance, which paves the way to a better understanding of how neural networks generalize across unseen domains.

**Our contributions.** We hypothesize that the model requires a gradient step of large magnitude when it fails to generalize well on test data from unseen domains. To quantify the magnitude of gradients, we propose a simple yet efficient gradient-based statistic, *GDSCORE*, which employs the norm of the gradients backpropagated from a standard cross-entropy loss on the test samples. To avoid the need for ground-truth labels, we propose a pseudo-labeling strategy that benefits from both correct and incorrect predictions. We demonstrate that the norm of this one-step gradient of the classification layer strongly correlates with the generalization performance under diverse distribution shifts, acting as a strong and lightweight proxy for the latter. The main contributions of our paper are summarized as follows:

1. We first provide several theoretical insights showing that correct pseudo-labeling and gradient norm have a direct impact on the test error estimation. This is achieved by looking at the analytical expression of the gradient after one backpropagation step over the pre-trained model on test data under distribution shift and by upper-bounding the target out-of-distribution risk.

2. Based on these theoretical insights, we propose the GDSCORE, which gauges the magnitude of the classification-layer gradients and presents a strong correlation with test accuracy. Our method does

      not require access to either test labels or training datasets and only needs one step of backpropagation which makes it particularly lightweight in terms of computational efficiency compared to other self-training methods.

3. We demonstrate the superiority of GdScore with a large-scale empirical evaluation. We achieve new state-of-the-art results on 11 benchmarks across diverse distribution shifts compared to 8 competitors, while being faster than the previous best baseline.

**Organization of the paper.** The rest of the paper is organized as follows. Section 3 presents the necessary background on the problem at hand. In Section 4, we derive the theoretical insights that motivate the GdScore introduced afterward. Section 5 is devoted to extensive empirical evaluation of our method, while the ablation study is deferred to Section 6. Finally, Section 8 concludes our work.

## 2 Related Work

**Unsupervised accuracy estimation.** Unsupervised accuracy estimation is a vital topic in practical applications due to frequent distribution shifts and the unavailability of ground-truth labels for test samples. To comprehensively understand this field, we introduce two main existing settings related to this topic.

1. Some works aim to estimate the test accuracy or gauge the accuracy discrepancy between the training and the test set only via the training data (Corneanu et al., 2020; Jiang et al., 2019; Neyshabur et al., 2017; Unterthiner et al., 2020; Yak et al., 2019; Martin & Mahoney, 2020). For example, the model-architecture-based algorithm (Corneanu et al., 2020) derives plenty of persistent topology properties from the training data, which can identify when the model learns to generalize to unseen datasets. However, those algorithms are deployed under the assumption that the training and the test data are from the same distribution, which means they are vulnerable to distribution shifts.

2. Our work belongs to the second setting, which aims to estimate the classification accuracy of a specific test dataset during evaluation using unlabeled test samples and/or labeled training datasets. The main research direction is to explore the negative relationship between the distribution discrepancy and model performance from the space of features (Deng & Zheng, 2021), parameters (Yu et al., 2022b) and labels (Lu et al., 2023). Another popular direction is to design an estimation score via the softmax outputs of the test samples (Guillory et al., 2021; Jiang et al., 2021; Guillory et al., 2021; Garg et al., 2022), which heavily relies on model calibration. Some works also learn from the field of unsupervised learning, such as agreement across multiple classifiers (Jiang et al., 2021; Madani et al., 2004; Platanios et al., 2016; 2017) and image rotation (Deng et al., 2021). In addition, the property of the test datasets presented during evaluation has also been studied recently (Xie et al., 2023). To the best of our knowledge, our work is the first to study the linear relationship between the gradients and model performance.

**Gradients in generalization.** The role of gradients in generalization has attracted increasing attention recently. To gauge the generalization performance of the hypothesis learned from the training data on unseen samples, known as the out-of-sample error (Hardt et al., 2016; London, 2017; Rivasplata et al., 2018), many studies try to provide a tight upper bound for generalization error from the view of gradient descent theoretically, indicating that gradients correlate with the discrepancy between the empirical loss and the population loss (Li et al., 2019; Chatterjee, 2020; Negrea et al., 2019; An et al., 2020). However, those works assume that seen to unseen data are from the identical distribution, while unsupervised accuracy estimation discusses a more complex and realistic issue that they come from different distributions. Under distribution shift, gradients are also explored. For example, Mansilla et al. (2021) clips the conflicting gradients emerging in domain shift scenarios and promotes gradient agreement across multiple domains via the gradient agreement strategy. Similarly, Zhao et al. (2022) designs a gradient-based regularization term to make the optimizer find the flat minima. In out-of-distribution (OOD) detection which goal is to determine whether a given sample is in-distribution (ID) or out-of-distribution (Hendrycks & Gimpel, 2016; Hendrycks et al., 2018; Liu et al., 2020; Yang et al., 2021; Liang et al., 2017), (Huang et al., 2021) finds that ID data usually have higher

gradient magnitude than OOD data from current source distribution to a uniform distribution. However, despite their empirical success, the relationship between gradients and generalization is still unclear.

## 3 Background

**Problem setup.** We consider a $K$-class classification task with the input space $\mathcal{X} \subset \mathbb{R}^D$ and the label set $\mathcal{Y} = \{1, \ldots, K\}$. Our learning model is a neural network with trainable parameters $\boldsymbol{\theta} \in \mathbb{R}^p$ that maps from the input space to the label space $f_{\boldsymbol{\theta}} : \mathcal{X} \to \mathbb{R}^K$. We view the network as a combination of a complex feature extractor $f_{\mathbf{g}}$ and a linear classification layer $f_{\boldsymbol{\omega}}$, where $\mathbf{g}$ and $\boldsymbol{\omega} = (\mathbf{w}_k)_{k=1}^K$ denote their corresponding parameters. Given a training example $\mathbf{x}_i$, the feedforward process can be expressed as:

$$f_{\boldsymbol{\theta}}(\mathbf{x}_i) = f_{\boldsymbol{\omega}}(f_{\mathbf{g}}(\mathbf{x}_i)). \tag{1}$$

Let $\mathbf{y} = (y^{(k)})_{k=1}^K$ denote the one-hot encoded vector of label $y$, i.e., $y^{(k)} = 1$ if and only if $y = k$, otherwise $y^{(k)} = 0$. Then, given a training dataset $\mathcal{D} = \{\mathbf{x}_i, y_i\}_{i=1}^n$ that consists of $n$ data points sampled *i.i.d.* from the source distribution $P_S(\mathbf{x}, y)$ defined over $\mathcal{X} \times \mathcal{Y}$, $f_{\boldsymbol{\theta}}$ is trained following the empirical cross-entropy loss minimization:

$$\mathcal{L}_{\mathcal{D}}(f_{\boldsymbol{\theta}}) = -\frac{1}{n} \sum_{i=1}^n \sum_{k=1}^K y_i^{(k)} \log \mathrm{s}_{\boldsymbol{\omega}}^{(k)}(f_{\mathbf{g}}(\mathbf{x}_i)), \tag{2}$$

where $\mathrm{s}_{\boldsymbol{\omega}}^{(k)}$ denotes the output of the softmax for the class $k$ approximating the posterior probability $P(Y = k|\mathbf{x})$, i.e., $\mathrm{s}_{\boldsymbol{\omega}}^{(k)}(f_{\mathbf{g}}(\mathbf{x})) = \exp\{\mathbf{w}_k^{\mathsf{T}} f_{\mathbf{g}}(\mathbf{x})\} / \left( \sum_{\tilde{k}} \exp\{\mathbf{w}_{\tilde{k}}^{\mathsf{T}} f_{\mathbf{g}}(\mathbf{x})\} \right)$.

**Unsupervised accuracy estimation.** We now assume to have access to $m$ test samples from the target distribution $\mathcal{D}_{\text{test}} = \{\tilde{\mathbf{x}}_i\}_{i=1}^m \sim P_T(\mathbf{x})$, where $P_T(\mathbf{x}, y) \neq P_S(\mathbf{x}, y)$. For each test sample $\tilde{\mathbf{x}}_i$, we predict the label by $\tilde{y}_i' = \arg\max_{k \in \mathcal{Y}} f_{\boldsymbol{\theta}}(\tilde{\mathbf{x}}_i)$. We now want to assess the performance of $f_{\boldsymbol{\theta}}$ on a target distribution without using corresponding ground-truth labels $\{\tilde{y}_i\}_{i=1}^m$ by estimating as accurately as possible the following quantity:

$$\mathrm{Acc}(\mathcal{D}_{\text{test}}) = \frac{1}{m} \sum_{i=1}^m \mathbb{1}(\tilde{y}_i' = \tilde{y}_i), \tag{3}$$

where $\mathbb{1}(\cdot)$ denotes the indicator function. In practice, unsupervised accuracy estimation methods provide a proxy score $S(\mathcal{D}_{\text{test}})$ that should exhibit a linear correlation with $\mathrm{Acc}(\mathcal{D}_{\text{test}})$. The performance of such methods is measured using the coefficient of determination $R_2$ and the Spearman correlation coefficient $\rho$.

## 4 On the strong correlation between gradient norm and test accuracy

We start by deriving an analytical expression of the gradient obtained when fine-tuning a source pre-trained model on new test data. We further use the intuition derived from it to propose our test accuracy estimation score and justify its effectiveness through a more thorough theoretical analysis.

**A motivational example.** Below, we follow the setup considered by Denevi et al. (2019); Balcan et al. (2019); Arnold et al. (2021) to develop our intuition behind the importance of gradient norm in unsupervised accuracy estimation. Our main departure point for this analysis is to consider fine-tuning: a popular approach to adapting a pre-trained model to different labeled datasets is to update either all or just a fraction of its parameters using gradient descent on the new data. To this end, let us consider the following linear regression example, where the test data from unseen domains are distributed as $X \sim \mathcal{N}(0, \sigma_t^x)$, $(Y|X = x) \sim \mathcal{N}(\theta_t x, 1)$ parameterized by the optimal regressor $\theta_t \in \mathbb{R}$, while the data on which the model was trained is distributed as $X \sim \mathcal{N}(0, \sigma_s^x)$, $(Y|X = x) \sim \mathcal{N}(\theta_s x, 1)$ with $\theta_s \in \mathbb{R}$. Consider the least-square loss over the test distribution:

$$\mathcal{L}_T(c) = \frac{1}{2} \mathbb{E}_{P_T(x,y)} (y - cx)^2.$$

When we do not observe the target labels, one possible solution would be to analyze fine-tuning when using the source generator $(Y|X=x) \sim \mathcal{N}(\theta_s x, 1)$ for pseudo-labeling. Then, we obtain that

$$
\begin{aligned}
\frac{1}{2} \nabla_c \mathbb{E}_{P_T(x)} \mathbb{E}_{P_S(y|x)}[(y - cx)^2] &= \mathbb{E}_{P_T(x)} \mathbb{E}_{P_S(y|x)}[(y - cx)(-x)] \\
&= \mathbb{E}_{P_T(x)} \mathbb{E}_{P_S(y|x)}[cx^2 - xy] \\
&= (c - \theta_s)\sigma_t^x \\
&= ((c - \theta_t) + (\theta_t - \theta_s))\sigma_t^x.
\end{aligned}
$$

This derivation, albeit simplistic, suggests that the gradient over the target data correlates – modulo the variance of $x$ – with $(c - \theta_t)$, capturing how far we are from the optimal parameters of the target model, and $(\theta_s - \theta_t)$, that can be seen as a measure of dissimilarity between the distributions of the optimal source and target parameters. Intuitively, both these terms are important for predicting the test accuracy performance suggesting that the gradient itself can be a good proxy for the latter.

### 4.1 Proposed approach: GdScore

We now formally introduce our proposed score, termed GDSCORE, to estimate test accuracy in an unsupervised manner during evaluation. We start by recalling the backpropagation process of the pre-trained neural network $f_\theta$ from a cross-entropy loss and then describe how to leverage the gradient norm for the unsupervised accuracy estimation. The detailed algorithm can be found in Appendix B.

**Feedforward.** Similar to the feedforward in the pre-training process shown in Eq. 1, for any given test individual $\tilde{\mathbf{x}}_i$, we have:

$$
f_{\boldsymbol{\theta}}(\tilde{\mathbf{x}}_i) = f_{\boldsymbol{\omega}}(f_{\mathbf{g}}(\tilde{\mathbf{x}}_i)). \tag{4}
$$

As explained above, we do not observe the true labels of test data. We now detail our strategy for pseudo-labeling that allows us to obtain accurate and balanced proxies for test data labels based on accurate and potentially inaccurate model predictions.

**Label generation strategy.** Unconditionally generating pseudo-labels for test data under distribution shift exhibits an obvious drawback: we treat all the assigned pseudo-labels as correct predictions when calculating the loss, ignoring the fact that some examples are possibly mislabeled. Therefore, we propose the following confidence-based label-generation policy that outputs for every $\tilde{\mathbf{x}}_i \in \mathcal{D}_{\text{test}}$:

$$
\tilde{y}_i' = \begin{cases} \arg\max_k f_{\boldsymbol{\theta}}(\tilde{\mathbf{x}}_i), & \max_k \mathrm{s}_{\boldsymbol{\omega}}^{(k)}(f_{\mathbf{g}}(\tilde{\mathbf{x}}_i)) > \tau \\ \tilde{y}' \sim U[1, K], & \text{otherwise} \end{cases} \tag{5}
$$

where $\tau$ denotes the threshold value, and $U[1, k]$ denotes the discrete uniform distribution with outcomes $\{1, \ldots, K\}$. In a nutshell, we assign the predicted label to $\tilde{\mathbf{x}}_i$, when the prediction confidence is larger than a threshold while using a randomly sampled label from the label space otherwise. The detailed empirical evidence justifying this choice is shown in Section 6, and we discuss the choice of proper threshold $\tau$ in Appendix 6. From the theoretical point of view, our approach assumes that the classifier makes mistakes mostly on data with low prediction confidence, for which we deliberately assign noisy pseudo-labels. Feofanov et al. (2019) used a similar approach to derive an upper bound on the test error and proved its tightness in the case where the assumption is satisfied. We discuss this matter in more detail in Appendix E.

**Backpropagation.** To estimate our score, we calculate the gradients w.r.t. the weights of the classification layer $\boldsymbol{\omega}$ during the first epoch backpropagated over the standard cross-entropy loss defined by:

$$
\mathcal{L}_{\mathcal{D}_{\text{test}}}(f_{\boldsymbol{\theta}}) = -\frac{1}{m} \sum_{i=1}^{m} \sum_{k=1}^{K} \tilde{y}_i'^{(k)} \log \mathrm{s}_{\boldsymbol{\omega}}^{(k)}(f_{\mathbf{g}}(\tilde{\mathbf{x}}_i)), \tag{6}
$$

where each unlabeled instance $\tilde{\mathbf{x}}_i$ is pseudo-labeled following Eq. 5. Then, the gradient of the classification layer $\boldsymbol{\omega}$ is evaluated as follows:

$$\nabla_{\boldsymbol{\omega}} \mathcal{L}_{\mathcal{D}_{\text{test}}}(f_{\boldsymbol{\theta}}) = -\frac{1}{m} \sum_{i=1}^{m} \sum_{k=1}^{K} \nabla_{\boldsymbol{\omega}} \left( \tilde{y}_i^{\prime(k)} \log \mathrm{s}_{\boldsymbol{\omega}}^{(k)}(f_{\mathbf{g}}(\tilde{\mathbf{x}}_i)) \right). \tag{7}$$

Note that our method requires neither gradients of the whole parameter set of the pre-trained model nor iterative training. This makes it highly computationally efficient.

**GdScore.** Now, we can define GDSCORE using a vector norm of gradients of the last layer. The score is expressed as follows:

$$S(\mathcal{D}_{\text{test}}) = \|\nabla_{\boldsymbol{\omega}} \mathcal{L}_{\mathcal{D}_{\text{test}}}(f_{\boldsymbol{\theta}})\|_p, \tag{8}$$

where $\| \cdot \|_p$ denotes $L_p$-norm.

## 4.2 Theoretical analysis

In this section, we provide theoretical insights into our method. We first clarify the connection between the true target cross-entropy error and the norm of the gradients. Then, we show that the gradient norm is upper-bounded by a weighted sum of the norm of the inputs.

**Notations.** For the sake of simplicity, we assume the feature extractor $f_{\mathbf{g}}$ is fixed and, by abuse of notation, we use $\mathbf{x}$ instead of $f_{\mathbf{g}}(\mathbf{x})$. Reusing the notations introduced in Section 3, the true target cross-entropy error writes

$$\mathcal{L}_T(\boldsymbol{\omega}) = -\mathbb{E}_{P_T(\mathbf{x},y)} \sum_k y^{(k)} \log \mathrm{s}_{\boldsymbol{\omega}}^{(k)}(\mathbf{x}),$$

where $\boldsymbol{\omega} = (\mathbf{w}_k)_{k=1}^{K} \in \mathbb{R}^{D \times K}$ are the parameters of the linear classification layer $f_{\boldsymbol{\omega}}$. For the ease of notation, the gradient of $\mathcal{L}_T$ w.r.t $\boldsymbol{\omega}$ is denoted by $\nabla \mathcal{L}_T$.

The following theorem, whose proof we defer to Appendix H.1, makes the connection between the true risk and the $L_p$-norm of the gradient explicit.

**Theorem 4.1** (Connection between the true risk and the $L_p$-norm of the gradient). *Let $\mathbf{c} \in \mathbb{R}^{D \times K}$ and $\mathbf{c}' \in \mathbb{R}^{D \times K}$ be two linear classifiers. For any $p, q \geq 1$ such that $\frac{1}{p} + \frac{1}{q} = 1$, we have that*

$$|\mathcal{L}_T(\mathbf{c}') - \mathcal{L}_T(\mathbf{c})| \leq \max_{\boldsymbol{\omega} \in \{\mathbf{c}', \mathbf{c}\}} (\|\nabla \mathcal{L}_T(\boldsymbol{\omega})\|_p) \cdot \|\mathbf{c}' - \mathbf{c}\|_q.$$

The left-hand side here is the difference in terms of the true risks obtained for the same distribution with two different classifiers. The right-hand side shows how this difference is controlled by the maximum gradient norm over the two classifiers and a term capturing how far the two are apart.

In the context of the proposed approach, we want to know the true risk of the source classifier $\boldsymbol{\omega}_s$ on the test data and its change after one step of gradient descent. The following corollary applies Theorem 4.1 to characterize this exact case. The proof is deferred to Appendix H.2.

**Corollary 4.2** (Connection after one gradient update). *Let $\mathbf{c}$ be the classifier obtained from $\boldsymbol{\omega}_s$ after one gradient descent step, i.e., $\mathbf{c} = \boldsymbol{\omega}_s - \eta \cdot \nabla \mathcal{L}_T(\boldsymbol{\omega}_s)$ with $\eta \geq 0$. Then, when $\boldsymbol{\omega} \in \{\boldsymbol{\omega}_s, \mathbf{c}\}$, we have that*

$$|\mathcal{L}_T(\boldsymbol{\omega}_s) - \mathcal{L}_T(\mathbf{c})| \leq \eta \max_{\boldsymbol{\omega}} (\|\nabla \mathcal{L}_T(\boldsymbol{\omega})\|_p) \|\nabla \mathcal{L}_T(\boldsymbol{\omega}_s)\|_q.$$

Note that in this case $\mathcal{L}_T(\boldsymbol{\omega}_s)$ can be seen as a term providing the true test risk after pseudo-labeling it with the source classifier. This shows the importance of pseudo-labeling as it acts as a departure point for obtaining a meaningful estimate of the right-hand side. When the latter is meaningful, the gradient norm on the right-hand side controls it together with a magnitude that tells us how far we went after one step of backpropagation.

In the next theorem, we provide an upper bound on the $L_p$-norm of the gradient as a weighted sum of the $L_p$-norm of the inputs. The proof is deferred to Appendix H.3.

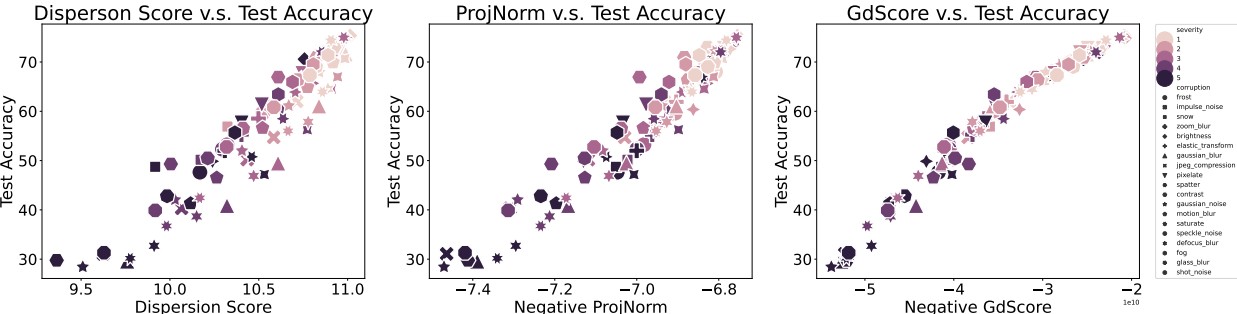

Figure 1: Test accuracy prediction versus True test accuracy on Entity-13 with ResNet18. We compare the performance of GDSCORE with that of Dispersion Score and ProjNorm via scatter plots. Each point represents one dataset under certain corruption and certain severity, where different shapes represent different types of corruption, and darker color represents the higher severity level.

**Theorem 4.3** (Upper-bounding the norm of the gradient). *For any $p \geq 1$ and $\forall \boldsymbol{\omega} \in \mathbb{R}^{D \times K}$, the $L_p$-norm of the gradient can be upper-bounded as follows:*

$$\|\nabla \mathcal{L}_T(\boldsymbol{\omega})\|_p \leq \mathbb{E}_{P_T(\mathbf{x}, y)} \alpha(\boldsymbol{\omega}, \mathbf{x}, y) \cdot \|\mathbf{x}\|_p \,,$$

*where $\alpha(\boldsymbol{\omega}, \mathbf{x}, y) = 1 - s_{\boldsymbol{\omega}}^{(k_y)}(\mathbf{x})$, with $k_y$ such that $y^{(k_y)} = 1$.*

Hence, the norm of the gradient is upper-bounded by a weighted combination of the norm of the inputs, where the weight $\alpha(\boldsymbol{\omega}, \mathbf{x}, y) \in [0, 1]$ conveys how well the model predicts on $\mathbf{x}$. In the case of perfect classification, the upper bound is tight and equals 0. In practice, as we do not have access to the true risk, the gradients can be approximated by the proposed GDSCORE that requires to pseudo-label test data by Eq. 5. As we said earlier, this implies that the model has to be well calibrated (see Appendix E), which is conventional to assume for self-training methods (Amini et al., 2023) including the approach of Yu et al. (2022b). Then, the network projects test data into the low confidence regions, and the gradient for these examples will be large as we need to update $\boldsymbol{\omega}_s$ significantly to fit them.

## 5 Experiments

### 5.1 Experimental setup

**Pre-training datasets.** For pre-training the neural network, we use CIFAR-10, CIFAR-100 (Krizhevsky & Hinton, 2009), TinyImageNet (Le & Yang, 2015), ImageNet (Deng et al., 2009), Office-31 (Saenko et al., 2010), Office-Home (Venkateswara et al., 2017), Camelyon17-WILDS (Koh et al., 2021), and BREEDS (Santurkar et al., 2020) which leverages class hierarchy of ImageNet (Deng et al., 2009) to create 4 datasets including Living-17, Nonliving-26, Entity-13 and Entity-30. In particular, to avoid time-consuming training, we directly utilize publicly available models pre-trained on Imagenet. For Office-31 and Office-Home, we train a neural network on every domain.

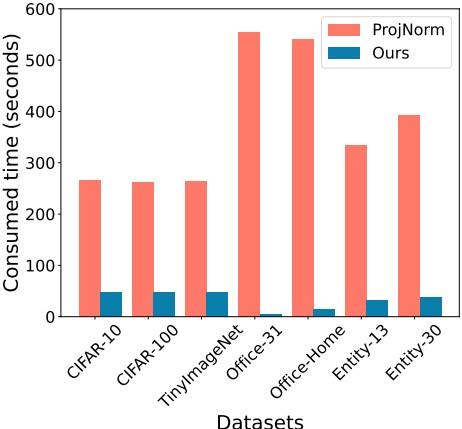

Figure 2: Runtime comparison of two self-training approaches with ResNet50.

**Test datasets.** In our comprehensive evaluation, we consider 11 datasets with 3 types of distribution shifts: synthetic, natural, and novel subpopulation shift. To verify the effectiveness of our method under the synthetic shift, we use CIFAR-10C, CIFAR-100C,

Table 1: Performance comparison on 11 benchmark datasets with ResNet18, ResNet50, and WRN-50-2, where $R^2$ refers to coefficients of determination, and $\rho$ refers to the absolute value of Spearman correlation coefficients (higher is better). The best results are highlighted in **bold**.

| Dataset | Network | Rotation $R^2$ | $\rho$ | ConfScore $R^2$ | $\rho$ | Entropy $R^2$ | $\rho$ | AgreeScore $R^2$ | $\rho$ | ATC $R^2$ | $\rho$ | Fréchet $R^2$ | $\rho$ | Dispersion $R^2$ | $\rho$ | ProjNorm $R^2$ | $\rho$ | Ours $R^2$ | $\rho$ |
|---|---|---|---|---|---|---|---|---|---|---|---|---|---|---|---|---|---|---|---|
| CIFAR 10 | ResNet18 | 0.822 | 0.951 | 0.869 | 0.985 | 0.899 | 0.987 | 0.663 | 0.929 | 0.884 | 0.985 | 0.950 | 0.971 | 0.968 | 0.990 | 0.936 | 0.982 | **0.971** | **0.994** |
| | ResNet50 | 0.835 | 0.961 | 0.935 | 0.993 | 0.945 | **0.994** | 0.835 | 0.985 | 0.946 | **0.994** | 0.858 | 0.964 | **0.987** | 0.990 | 0.944 | 0.989 | 0.969 | 0.993 |
| | WRN-50-2 | 0.862 | 0.976 | 0.943 | **0.994** | 0.942 | **0.994** | 0.856 | 0.986 | 0.947 | **0.994** | 0.814 | 0.973 | 0.962 | 0.988 | 0.961 | 0.989 | **0.971** | **0.994** |
| | Average | 0.840 | 0.963 | 0.916 | 0.991 | 0.930 | 0.992 | 0.785 | 0.967 | 0.926 | 0.991 | 0.874 | 0.970 | **0.972** | 0.990 | 0.947 | 0.987 | 0.970 | 0.994 |
| CIFAR 100 | ResNet18 | 0.860 | 0.936 | 0.916 | 0.985 | 0.891 | 0.979 | 0.902 | 0.973 | 0.938 | 0.986 | 0.888 | 0.968 | 0.952 | 0.988 | 0.979 | 0.980 | **0.987** | **0.996** |
| | ResNet50 | 0.908 | 0.962 | 0.919 | 0.984 | 0.884 | 0.977 | 0.922 | 0.982 | 0.921 | 0.984 | 0.837 | 0.972 | 0.951 | 0.985 | 0.988 | 0.991 | **0.991** | **0.997** |
| | WRN-50-2 | 0.924 | 0.970 | 0.971 | 0.984 | 0.968 | 0.981 | 0.955 | 0.977 | 0.978 | 0.993 | 0.865 | 0.987 | 0.980 | 0.991 | 0.990 | 0.991 | **0.995** | **0.998** |
| | Average | 0.898 | 0.956 | 0.936 | 0.987 | 0.915 | 0.983 | 0.927 | 0.982 | 0.946 | 0.988 | 0.864 | 0.976 | 0.962 | 0.988 | 0.985 | 0.987 | **0.991** | **0.997** |
| TinyImageNet | ResNet18 | 0.786 | 0.946 | 0.670 | 0.869 | 0.592 | 0.842 | 0.561 | 0.853 | 0.751 | 0.945 | 0.826 | 0.970 | 0.966 | 0.986 | 0.970 | 0.981 | **0.971** | **0.994** |
| | ResNet50 | 0.786 | 0.947 | 0.670 | 0.869 | 0.651 | 0.892 | 0.560 | 0.853 | 0.751 | 0.945 | 0.826 | 0.971 | 0.977 | 0.986 | 0.979 | 0.987 | **0.980** | **0.995** |
| | WRNt-50-2 | 0.878 | 0.967 | 0.757 | 0.951 | 0.704 | 0.935 | 0.654 | 0.904 | 0.635 | 0.897 | 0.884 | 0.984 | 0.968 | 0.986 | 0.965 | 0.983 | **0.975** | **0.996** |
| | Average | 0.805 | 0.959 | 0.727 | 0.920 | 0.650 | 0.890 | 0.599 | 0.878 | 0.693 | 0.921 | 0.847 | 0.976 | 0.970 | 0.987 | 0.972 | 0.984 | **0.976** | **0.995** |
| ImageNet | ResNet18 | - | - | 0.979 | 0.991 | 0.963 | 0.991 | - | - | 0.974 | 0.983 | 0.802 | 0.974 | 0.940 | 0.971 | 0.975 | 0.993 | **0.986** | **0.996** |
| | ResNet50 | - | - | 0.980 | 0.994 | 0.967 | 0.992 | - | - | 0.970 | 0.983 | 0.855 | 0.974 | 0.938 | 0.968 | 0.986 | 0.993 | **0.987** | **0.996** |
| | WRNt-50-2 | - | - | 0.983 | 0.991 | 0.963 | 0.991 | - | - | 0.983 | 0.993 | 0.909 | 0.988 | 0.939 | 0.976 | 0.978 | 0.993 | **0.984** | **0.998** |
| | Average | - | - | 0.981 | 0.993 | 0.969 | 0.992 | - | - | 0.976 | 0.987 | 0.855 | 0.979 | 0.939 | 0.972 | 0.980 | 0.993 | **0.986** | **0.997** |
| Office-31 | ResNet18 | **0.753** | **0.942** | 0.470 | 0.828 | 0.322 | 0.714 | 0.003 | 0.085 | 0.843 | 0.942 | 0.143 | 0.257 | 0.618 | 0.714 | 0.099 | 0.428 | 0.675 | 0.829 |
| | ResNet50 | 0.391 | 0.828 | 0.485 | 0.828 | 0.354 | 0.828 | 0.011 | 0.463 | 0.532 | 0.485 | 0.034 | 0.257 | 0.578 | 0.714 | 0.240 | 0.428 | **0.604** | **0.829** |
| | WRN-50-2 | 0.577 | 0.6 | 0.524 | 0.714 | 0.424 | 0.714 | 0.002 | 0.257 | 0.405 | 0.942 | 0.034 | 0.142 | **0.671** | 0.714 | 0.147 | 0.143 | 0.544 | **0.829** |
| | Average | 0.567 | 0.790 | 0.493 | 0.790 | 0.367 | 0.276 | 0.006 | 0.211 | 0.593 | 0.790 | 0.071 | 0.047 | **0.622** | 0.714 | 0.162 | 0.333 | 0.608 | **0.829** |
| Office-Home | ResNet18 | 0.822 | 0.930 | 0.795 | 0.909 | 0.761 | 0.881 | 0.054 | 0.146 | 0.571 | 0.615 | 0.605 | 0.755 | 0.453 | 0.664 | 0.064 | 0.202 | **0.876** | **0.909** |
| | ResNet50 | **0.851** | **0.944** | 0.769 | 0.895 | 0.742 | 0.853 | 0.026 | 0.216 | 0.487 | 0.734 | 0.607 | 0.685 | 0.383 | 0.727 | 0.169 | 0.475 | 0.829 | 0.944 |
| | WRN-50-2 | **0.823** | **0.958** | 0.741 | 0.874 | 0.696 | 0.846 | 0.132 | 0.405 | 0.383 | 0.643 | 0.589 | 0.706 | 0.456 | 0.713 | 0.172 | 0.531 | 0.809 | 0.916 |
| | Average | 0.832 | 0.944 | 0.768 | 0.892 | 0.733 | 0.860 | 0.071 | 0.256 | 0.480 | 0.664 | 0.601 | 0.715 | 0.431 | 0.702 | 0.135 | 0.403 | **0.837** | **0.923** |
| Camelyon17-WILDS | ResNet18 | 0.944 | **1.000** | 0.980 | **1.000** | 0.980 | **1.000** | 0.977 | **1.000** | 0.981 | **1.000** | 0.988 | **1.000** | 0.992 | **1.000** | 0.612 | 0.500 | **0.996** | **1.000** |
| | ResNet50 | 0.931 | **1.000** | 0.994 | **1.000** | 0.993 | **1.000** | 0.998 | **1.000** | 0.993 | **1.000** | 0.971 | **1.000** | 0.012 | 0.500 | 0.811 | **1.000** | **0.999** | **1.000** |
| | WRN-50-2 | 0.918 | **1.000** | 0.944 | **1.000** | 0.945 | **1.000** | 0.965 | **1.000** | 0.942 | **1.000** | 0.994 | **1.000** | 0.001 | 0.500 | 0.789 | 0.500 | **0.997** | **1.000** |
| | Average | 0.931 | **1.000** | 0.973 | **1.000** | 0.980 | **1.000** | 0.982 | **1.000** | 0.972 | **1.000** | 0.984 | **1.000** | 0.334 | 0.667 | 0.737 | 0.667 | **0.998** | **1.000** |
| Entity-13 | ResNet18 | 0.927 | 0.961 | 0.795 | 0.940 | 0.794 | 0.935 | 0.543 | 0.919 | 0.823 | 0.945 | 0.950 | 0.981 | 0.937 | 0.968 | 0.952 | 0.981 | **0.969** | **0.991** |
| | ResNet50 | 0.932 | 0.976 | 0.728 | 0.941 | 0.698 | 0.928 | 0.901 | 0.964 | 0.783 | 0.950 | 0.903 | 0.959 | 0.764 | 0.892 | 0.944 | 0.974 | **0.960** | **0.995** |
| | WRN-50-2 | 0.939 | 0.983 | 0.930 | 0.977 | 0.919 | 0.973 | 0.871 | 0.935 | 0.936 | 0.980 | 0.906 | 0.958 | 0.815 | 0.905 | 0.950 | 0.977 | **0.968** | **0.995** |
| | Average | 0.933 | 0.973 | 0.817 | 0.953 | 0.804 | 0.945 | 772 | 0.939 | 0.847 | 0.958 | 0.920 | 0.966 | 0.948 | 0.977 | 0.839 | 0.922 | **0.966** | **0.994** |
| Entity-30 | ResNet18 | 0.964 | 0.979 | 0.570 | 0.836 | 0.553 | 0.832 | 0.542 | 0.935 | 0.611 | 0.845 | 0.849 | 0.978 | 0.929 | 0.968 | 0.952 | 0.987 | **0.970** | **0.995** |
| | ResNet50 | **0.961** | 0.980 | 0.878 | 0.969 | 0.838 | 0.956 | 0.914 | 0.975 | 0.924 | 0.973 | 0.835 | 0.956 | 0.783 | 0.914 | 0.937 | 0.986 | 0.957 | **0.996** |
| | WRN-50-2 | 0.940 | 0.978 | 0.897 | 0.974 | 0.756 | 0.970 | 0.826 | 0.955 | 0.936 | 0.984 | 0.927 | 0.973 | 0.927 | 0.973 | **0.959** | 0.986 | 0.949 | **0.994** |
| | Average | 0.955 | 0.978 | 0.781 | 0.926 | 0.756 | 0.919 | 0.728 | 0.956 | 0.823 | 0.934 | 0.871 | 0.969 | 0.880 | 0.952 | 0.949 | 0.987 | **0.959** | **0.995** |
| Living-17 | ResNet18 | 0.876 | 0.973 | 0.913 | 0.973 | 0.898 | 0.970 | 0.586 | 0.736 | 0.940 | 0.973 | 0.768 | 0.950 | 0.900 | 0.958 | 0.923 | 0.970 | **0.949** | **0.983** |
| | ResNet50 | 0.906 | 0.956 | 0.880 | 0.967 | 0.853 | 0.961 | 0.633 | 0.802 | **0.938** | **0.976** | 0.771 | 0.926 | 0.851 | 0.929 | 0.903 | 0.924 | 0.931 | 0.975 |
| | WRN-50-2 | 0.909 | 0.957 | 0.928 | 0.980 | 0.921 | 0.977 | 0.652 | 0.793 | **0.966** | **0.984** | 0.931 | 0.967 | 0.931 | 0.966 | 0.915 | 0.970 | 0.910 | 0.976 |
| | Average | 0.933 | 0.974 | 0.907 | 0.973 | 0.814 | 0.969 | 0.623 | 0.777 | **0.948** | **0.978** | 0.817 | 0.949 | 0.894 | 0.951 | 0.913 | 0.969 | 0.930 | **0.978** |
| Nonliving-26 | ResNet18 | 0.906 | 0.955 | 0.781 | 0.925 | 0.739 | 0.909 | 0.543 | 0.810 | 0.854 | 0.939 | 0.914 | 0.980 | **0.958** | 0.981 | 0.939 | 0.978 | 0.953 | **0.983** |
| | ResNet50 | 0.916 | 0.970 | 0.832 | 0.942 | 0.776 | 0.918 | 0.638 | 0.837 | 0.893 | 0.960 | 0.848 | 0.950 | 0.805 | 0.907 | 0.873 | 0.972 | **0.945** | **0.989** |
| | WRN-50-2 | 0.917 | 0.977 | 0.932 | 0.971 | 0.912 | 0.959 | 0.676 | 0.861 | **0.945** | 0.969 | 0.885 | 0.942 | 0.893 | 0.939 | 0.924 | 0.973 | 0.937 | **0.985** |
| | Average | 0.913 | 0.967 | 0.849 | 0.946 | 0.809 | 0.929 | 0.618 | 0.836 | 0.897 | 0.956 | 0.882 | 0.957 | 0.913 | 0.974 | 0.886 | 0.943 | **0.945** | **0.985** |

and ImageNet-C (Hendrycks & Dietterich, 2019) that span 19 types of corruption across 5 severity levels, as well as TinyImageNet-C (Hendrycks & Dietterich, 2019) with 15 types of corruption and 5 severity levels. For the natural shift, we use the domains excluded from training from Office-31, Office-Home, and Camelyon17-WILDS as the OOD datasets. For the novel subpopulation shift, we consider the BREEDS benchmarks, namely, Living-17, Nonliving-26, Entity-13, and Entity-30, which are constructed from ImageNet-C.

**Training details.** To show the versatility of our approach across different architectures, we perform all our experiments on ResNet18, ResNet50 (He et al., 2016) and WRN-50-2 (Zagoruyko & Komodakis, 2016) models. We train them for 20 epochs for CIFAR-10 (Krizhevsky & Hinton, 2009) and 50 epochs for the other datasets. In all cases, we use SGD with a learning rate of $10^{-3}$, cosine learning rate decay (Loshchilov & Hutter, 2016), a momentum of 0.9, and a batch size of 128. For all experiments, we use $p = 0.3$ to compute GDSCORE.

**Evaluation metrics.** We measure the performance of all competing methods using the coefficients of determination ($R^2$) and Spearman correlation coefficients ($\rho$) calculated between the baseline scores and the

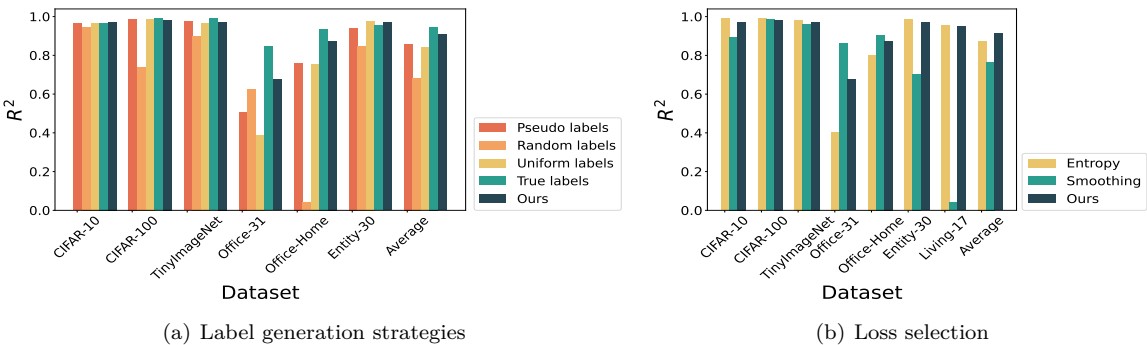

(a) Label generation strategies          (b) Loss selection

Figure 3: Performance comparison ($R^2$) on 7 datasets with ResNet18 between (a) different label generation strategies and (b) different types of losses across 3 types of distribution shifts. Results confirm that our proposed method performs better on average across various datasets and types of shifts.

true test error. To compare the computational efficiency with two self-training methods, we calculate the average evaluation time needed for every test dataset.

**Baselines.** We compare our method GDSCORE with 8 baselines commonly considered in the unsupervised accuracy estimation literature: *Rotation Prediction* (Rotation) (Deng et al., 2021), *Averaged Confidence* (ConfScore) (Hendrycks & Gimpel, 2016), *Entropy* (Guillory et al., 2021), *Agreement Score* (AgreeScore) (Jiang et al., 2021), *Averaged Threshold Confidence* (ATC) (Garg et al., 2022), *AutoEval* (Fréchet) (Deng & Zheng, 2021), *Dispersion Score* (Dispersion) (Xie et al., 2023), and *ProjNorm* (Yu et al., 2022b). The first six methods are training-free and the last method is an instance of self-training approaches. More details about the baselines can be found in Appendix C.

## 5.2 Main takeaways

**GdScore correlates with test accuracy stronger than baselines across diverse distribution shifts.** In Table 1, we present the OOD error estimation performance on 11 benchmark datasets across 3 model architectures as measured by $R^2$ and $\rho$. We observe that GDSCORE outperforms existing methods under diverse distribution shifts. Our method achieves an average $R^2$ higher than 0.99 on CIFAR-100, while the average $R^2$ of the other baselines is always below. In addition, our method performs stably across different distribution shifts compared with the other existing algorithms. For example, Rotation performs well under the natural shift but experiences a dramatic performance drop under the synthetic shift, ranking from the second best to the eighth. However, our method achieves consistently high performance, ranking the best on average across the three types of distribution shifts.

Furthermore, we provide the visualization of estimation performance in Fig. 1, where we present the scatter plots for Dispersion Score, ProjNorm and GDSCORE on Entity-13 with ResNet18. We can see that GDSCORE and test accuracy have a strong linear relationship, while the other state-of-the-art methods struggle to have a linear correlation in cases when the test error is high. This phenomenon demonstrates the superiority of GDSCORE in unsupervised accuracy estimation. In the next paragraph, we also demonstrate the computational efficiency of our approach.

**GdScore is a more efficient self-training approach.** In the list of baselines, both our method and ProjNorm (Yu et al., 2022b) belong to self-training methods (Amini et al., 2023). The latter, however, requires costly iterative training on the neural network during evaluation to obtain the complete set of fine-tuned parameters to calculate the distribution discrepancy in network parameters. Compared with ProjNorm, our method only trains the model for one epoch and collects the gradients of the linear classification layer to calculate the gradient norm, which is much more computationally efficient. Fig. 2 presents the comparison of computational efficiency between the two methods on 7 datasets with ResNet50. From this figure, we can see that our method is up to 80% faster than ProjNorm on average. This difference is striking on the Office-31

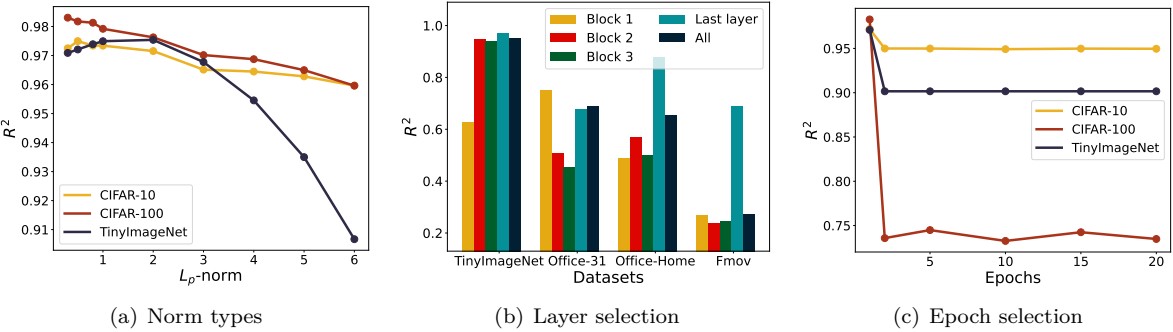

(a) Norm types        (b) Layer selection        (c) Epoch selection

Figure 5: Sensitivity analysis on the effect of (a) norm types, (b) layer selection for gradients, and (c) epoch selection. The first and the third experiments are conducted on CIFAR-10C, CIFAR-100C, and TinyImageNet-C, while the second experiment includes TinyImageNet-C, Office-31, Office-Home, and WILDS-FMoV (Koh et al., 2021). All experiments are conducted with ResNet18.

dataset, where our method is not only two orders of magnitude faster than ProjNorm but also improves the $R^2$ score by a factor of 2.

**Robustness of our approach.** GDSCORE achieves great performance across all datasets, architectures, and types of shifts (Table 1). To highlight the robustness of our approach, we compare the distributions of $R^2$ in Figure 4, including as additional baseline the very recent *Nuclear Norm* (Deng et al., 2023) (more results in Appendix F). We show that GDSCORE is the best and most stable approach on 10 datasets (except ImageNet).

# 6 Ablation study

In this section, we will conduct comprehensive ablation studies to verify the efficiency of our proposed label-generation strategy, and choice of hyperparameters such as $\tau$ and $p$ as well as self-training settings to obtain GdScore such as epochs, blocks, and losses.

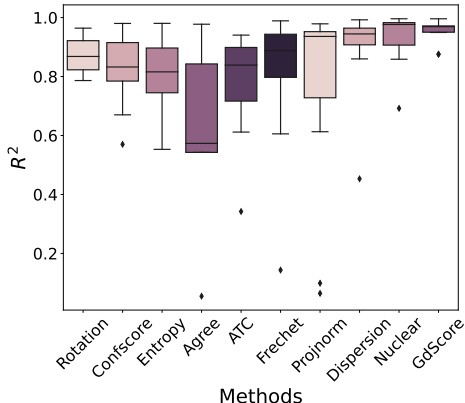

Figure 4: Robustness comparison for all estimation baselines across diverse distribution shifts with ResNet18.

**Random pseudo-labels boost the performance under natural shift.** In Fig. 3(a), we conduct an ablation study to verify the effectiveness of our label generation strategy by comparing it with ground-truth labels, uniform labels (Huang et al., 2021), full random labels, and full pseudo-labeling. From the figure, we observe that while comparable with the ground truth under the synthetic drift and the subpopulation shift, the other strategies lead to a drastic drop in performance under the natural shift. This phenomenon is possibly caused by imprecise gradient calculation based on incorrect pseudo labels. Our labeling strategy performs better on average suggesting that random labels for low-confidence samples provide certain robustness of the score under natural shift.

**Cross-entropy loss is robust to different shifts.** To demonstrate the impact of different losses on the test accuracy estimation performance, we compare the standard cross-entropy loss used in our method with the entropy loss for samples with low confidence (see definition in Appendix D). Moreover, we verify the effectiveness of the label smoothing, a simple yet effective tool for model calibration and performance improvement (Müller et al., 2019), by setting the smoothing rate as 0.4.

Fig. 3(b) illustrates this comparison revealing that standard cross-entropy loss is the most robust choice across different types of distribution shifts. We also note that the entropy loss enhances the estimation

performance under the synthetic and the novel subpopulation shifts, but struggles under the natural shift. On the contrary, the label smoothing regularization can increase the performance under the natural shift but decrease it under the synthetic and the novel subpopulation shifts.

**Choosing smaller $p$ for better estimation performance.** To illustrate the effect of the choice of $L_p$-norm on the performance, we conduct a sensitivity analysis on three datasets with ResNet18 summarized in Figure 5(a). We can see that there is an obvious decreasing trend as $p$ becomes larger. Especially, when $p$ is smaller than 1, the estimation performance can fluctuate within a satisfying range. This is probably because a smaller $p$ (i.e., $0 < p < 1$) makes the $L_p$-norm more suitable for the high-dimensional space, while the $L_p$-norm with $p \geq 1$ is likely to ignore gradients that are close to 0 (Wang et al., 2016; Huang et al., 2021), so for all experiments, we used $p = 0.3$. The remark below, whose proof we defer to Appendix H.4, provides some theoretical insights into the $L_p$-norm of the gradient for $0 < p < 1$.

**Remark 6.1** (Case $0 < p < 1$). *Let $\mathbf{c}$ be the classifier obtained from $\boldsymbol{\omega}_s$ after one gradient descent step, i.e., $\mathbf{c} = \boldsymbol{\omega}_s - \eta \cdot \nabla \mathcal{L}_T(\boldsymbol{\omega}_s)$ with $\eta \geq 0$. For any $p \in (0,1)$, we have*

$$\eta \|\nabla \mathcal{L}_T(\boldsymbol{\omega}_s)\|_p \leq |\|\mathbf{c}\|_p - \|\boldsymbol{\omega}_s\|_p|.$$

**Gradients from the last layer can provide sufficient information.** In this part, we aim to understand whether backpropagation through other layers in the neural network can provide a better test accuracy estimate. For this, we separate the feature extractor $f_{\mathbf{g}}$ into 3 blocks of layers with roughly equal size and calculate the GRDNORM scores for each of them. Additionally, we also try to gather the gradients over the whole network. Fig. 5(b) plots the obtained results on 4 datasets, suggesting that the last layer provides sufficient information to predict the true test accuracy in an unsupervised way.

**No gains after 1 epoch of backpropagation.** Here, we train the neural network for $r$ epochs, where $r \in \{1, 2, 5, 10, 15, 20\}$, and store the gradient vectors of the classification layer for each value of $r$. Fig. 5(c) suggests that the gradient norms after 1 step are sufficient to predict the model performance under distribution shifts. Further training gradually degrades the performance with the increasing $r$. The reason behind the phenomenon is that the gradients in the first epoch contain the most abundant information about the training dataset, while the neural network fine-tuned on the test dataset for several epochs is going to forget previous training categories (Kemker et al., 2018).

**Choice of proper threshold $\tau$** In our experiments (see Section 5), we set the value of $\tau$ as 0.5 across all datasets and network architectures. This choice of $\tau$ is due to the intuition that if a label contains a softmax probability below 0.5, it means that this predicted label has over 50% chances of being wrong. It means that this label has a higher probability of being incorrect than to be correct. Thus, we tend to regard it as an incorrect prediction. To demonstrate the impact of threshold $\tau$ on the final performance, we conduct an ablation study on CIFAR-10C and Office-31 with ResNet18 using varying values of $\tau$. We display in Table 2 the corresponding values of $R^2$. We can observe that the final performance improves and achieves its best value for $\tau$ is 0.5, before decreasing slightly.

Table 2: Performance on CIFAR-10 and Office-31 with ResNet18 for varying value of $\tau$. The metric used in this table is the coefficient of determination $R^2$. The best result is highlighted in **bold**.

| Threshold | 0.0 | 0.1 | 0.2 | 0.3 | 0.4 | 0.5 | 0.6 | 0.7 | 0.8 | 0.9 |
|---|---|---|---|---|---|---|---|---|---|---|
| CIFAR-10C | 0.963 | 0.963 | 0.964 | 0.965 | 0.971 | **0.972** | 0.967 | 0.962 | 0.963 | 0.959 |
| Office-31 | 0.495 | 0.498 | 0.532 | 0.674 | **0.685** | 0.667 | 0.545 | 0.451 | 0.114 | 0.131 |

## 7 Connection to GradNorm

A current work, GradNorm (Huang et al., 2021), employs gradients to detect OOD samples whose labels belong to a different label space from the training data. It gauges the magnitude of gradients in the classification

layer, backpropagated from a KL divergence between the softmax probability and uniform distribution. Compared with GradNorm, our method bears three critical differences, in terms of the problem setting, methodology, and theoretical insights.

1) *Problem setting*: GradNorm focuses on OOD detection, which aims to determine whether a given sample is in-distribution (ID) or out-of-distribution (Hendrycks & Gimpel, 2016; Hendrycks et al., 2018; Liu et al., 2020; Yang et al., 2021; Liang et al., 2017), while our method aims to estimate the test accuracy without ground-truth test labels. It requires GradNorm should be an instance-level score for classification, but our method is a dataset-level score for linear regression. Furthermore, in OOD detection, the label spaces of OOD data and training data are disjoint, while in unsupervised test accuracy, the training and the OOD label spaces are shared. The two are also evaluated differently: AUROC score for OOD detection and correlation coefficients for error estimation. Those differences are summarized in Table 3.

Table 3: Main differences between OOD detection and OOD error estimation.

| Learning Problem | Goal | Scope | Metric |
|---|---|---|---|
| OOD detection | Predict ID/OOD | $\tilde{\mathbf{x}}_i$ | AUROC |
| OOD error estimation | Proxy to test error | $\mathcal{D}_{\text{test}}$ | $R^2$ and $\rho$ |

2) *Methodology*: GradNorm obtains the magnitude of gradients via a KL-divergence loss measuring the distribution distance from the training distribution to a uniform distribution, while our method obtains them using a standard cross entropy loss with the specifically designed label generation strategy. GradNorm assumes that OOD data should have a lower magnitude of gradients, but in our cases, test data are certified to have a higher magnitude of gradients. Table 4 presents the performance comparison of the two methods in unsupervised accuracy estimation on 7 datasets across 3 types of distribution shifts with ResNet18. It illustrates that GradNorm is inferior to our approach for unsupervised accuracy estimation suggesting that the two problems cannot be tackled with the same tools.

3) *Theoretical insights*: GradNorm is certified to capture the joint information between features and outputs to detect OOD data from the oncoming dataset. However, the reason why OOD data have a lower magnitude of gradients is still unclear. Our method provides clearer theoretical insights to demonstrate the relationship between gradients and test accuracy even under distribution shift, which further inspires future work to address generalization issues from the view of gradients

Table 4: Performance comparison between Huang et al. (2021) and our methods on 7 datasets with ResNet18. The metric used in this table is the coefficient of determination $R^2$. The best results are highlighted in **bold**.

| Method | CIFAR 10 | CIFAR 100 | TinyImageNet | Office-31 | Office-Home | Entity-30 | Living-17 |
|---|---|---|---|---|---|---|---|
| (Huang et al., 2021) | 0.951 | 0.978 | 0.894 | 0.596 | 0.848 | 0.964 | 0.942 |
| Ours | **0.972** | **0.983** | **0.971** | **0.675** | **0.876** | **0.970** | **0.949** |

## 8 Conclusion

In this paper, we showcased the strong linear relationship between the magnitude of gradients and the model performance under distribution shifts. We proposed GDSCORE, a simple yet efficient method to estimate the ground-truth test accuracy error by measuring the gradient magnitude of the last classification layer. Our method consistently achieves superior performance across various distribution shifts than previous works. Furthermore, it does not require the detailed architecture of the feature extractors and training datasets. Those properties guarantee that our method can be easily deployed in the real world, and meet practical demands, such as large models and confidential information. We hope that our research sheds new light on the usefulness of gradient norms for unsupervised accuracy estimation.

**Acknowledgments**

This research is supported by the National Research Foundation Singapore and DSO National Laboratories under the AI Singapore Programme (AISGAward No: AISG2-GC-2023-009).

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

## A Appendix

## B Pseudo-code of GdScore

Our proposed GDSCORE for unsupervised accuracy estimation can be calculated as shown in Algorithm 1.

---

**Algorithm 1:** Unsupervised Accuracy Estimation via GDSCORE

---

**Input:** Test dataset from unseen domains $\tilde{\mathcal{D}} = \{\tilde{\mathbf{x}}_i\}_{i=1}^m$, a pre-trained model $f_{\boldsymbol{\theta}} = f_{\mathbf{g}} \circ f_{\boldsymbol{\omega}}$ (feature extractor $f_{\mathbf{g}}$ and classifier $f_{\boldsymbol{\omega}}$), a threshold value $\tau$.
**Output:** The GDSCORE.
**for** each test instance $\tilde{\mathbf{x}}_i$ **do**
    Obtain the maximum softmax probability via $\tilde{r}_i = \max_k s_{\boldsymbol{\omega}}^{(k)}(f_{\mathbf{g}}(\tilde{\mathbf{x}}_i))$ .
    **if** $\tilde{r}_i > \tau$ **then**
        Obtain pseudo labels via $\tilde{y}_i' = \arg\max_k f_{\boldsymbol{\theta}}(\tilde{\mathbf{x}}_i)$,
    **else**
        Obtain random labels via $\tilde{y}_i' \sim U[1, K]$.
    **end if**
**end for**
Calculate the cross-entropy loss using assigned labels $\tilde{y}_i$ via Eq. 6.
Calculate gradients of the weights in the classification layer via Eq. 7.
Calculate GDSCORE $S(\tilde{\mathcal{D}})$ via Eq. 8.

---

## C Baselines

**Rotation.** (Deng et al., 2021) By rotating images from both the training and the test sets with different angles, we can obtain new inputs and their corresponding labels $y_i^r$ which indicate by how many degrees they rotate. During pre-training, an additional classifier about rotation degrees should be learned. Then the *Rotation Prediction* (Rotation) metric can be calculated as:

$$S_r(\mathcal{D}_{\text{test}}) = \frac{1}{m}\sum_{i=1}^m \Big(\frac{1}{4}\sum_{r\in\{0°,90°,180°,270°\}}(\mathbb{1}(\hat{y}_i^r \neq y_i^r))\Big),$$

where $\hat{y}_i^r$ denotes the predicted labels about rotation degrees.

**ConfScore.** (Hendrycks & Gimpel, 2016) This metric directly leverages the average maximum softmax probability as the estimation of the test error, which is expressed as:

$$S_{cf}(\mathcal{D}_{\text{test}}) = \frac{1}{m}\sum_{i=1}^m \max(s_{\boldsymbol{\omega}}(f_{\boldsymbol{g}}(\tilde{\mathbf{x}}_i))).$$

**Entropy.** (Guillory et al., 2021) This metric estimates the test error via the average entropy loss:

$$S_e(\mathcal{D}_{\text{test}}) = \frac{1}{m}\sum_{i=1}^m\sum_{k=1}^K s_{\boldsymbol{\omega}}^{(k)}(f_{\boldsymbol{g}}(\tilde{\mathbf{x}}_i)) \log s_{\boldsymbol{\omega}}^{(k)}(f_{\boldsymbol{g}}(\tilde{\mathbf{x}}_i)).$$

**AgreeScore.** (Jiang et al., 2021) This method trains two independent neural networks simultaneously during pre-training, and estimates the test error via the rate of disagreement across the two models:

$$S_{ag}(\mathcal{D}_{\text{test}}) = \frac{1}{m}\sum_{i=1}^m \mathbb{1}(\tilde{y}_{1,i}' \neq \tilde{y}_{2,i}'),$$

where $\tilde{y}_{1,i}'$ and $\tilde{y}_{2,i}'$ denote the predicted labels by the two models respectively.

**ATC.** (Garg et al., 2022) It measures how many test samples have a confidence larger than a threshold that is learned from the source distribution. It can be expressed as:

$$S_{atc}(\mathcal{D}_{\text{test}}) = \frac{1}{m} \sum_{i=1}^{m} \mathbb{1}(\sum_{k=1}^{K} s_{\boldsymbol{\omega}}^{(k)}(f_{\boldsymbol{g}}(\tilde{\mathbf{x}}_i)) \log s_{\boldsymbol{\omega}}^{(k)}(f_{\boldsymbol{g}}(\tilde{\mathbf{x}}_i)) < t),$$

where $t$ is the threshold value learned from the validation set of the training dataset.

**Fréchet.** (Deng & Zheng, 2021) This method utilizes Fréchet distance to measure the distribution gap between the training and the test datasets, which serves the test error estimation:

$$S_{fr}(\mathcal{D}_{\text{test}}) = ||\mu_{train} - \mu_{test}|| + Tr(\Sigma_{train} + \Sigma_{test} - 2(\Sigma_{train}\Sigma_{test})^{\frac{1}{2}}),$$

where $\mu_{train}$ and $\mu_{test}$ denote the mean feature vector of $\mathcal{D}$ and $\mathcal{D}_{test}$, respectively. $\Sigma_{train}$ and $\Sigma_{test}$ refer to the covariance matrices of corresponding datasets.

**Dispersion.** (Xie et al., 2023) This paper estimates the test error by gauging the feature separability of the test dataset in the feature space:

$$S_{dis}(\mathcal{D}_{\text{test}}) = \log \frac{\sum_{k=1}^{K} m_k \cdot ||\bar{\boldsymbol{\mu}} - \tilde{\boldsymbol{\mu}}_k||_2^2}{K - 1},$$

where $\boldsymbol{\mu}$ denotes the center of the whole features, and $\boldsymbol{\mu}_k$ denotes the mean of $k^{th}$-class features.

**Nuclear Norm.** (Deng et al., 2023) This paper estimates the test error by computing the nuclear norm of final softmax probabilities, which can be expressed as:

$$S_{nu}(\mathcal{D}_{\text{test}}) = ||P||_*,$$

where Nuclear norm $||P||_*$ is defined as the sum of singular values of $P$.

**ProjNorm.** (Yu et al., 2022b) This method fine-tunes the pre-trained model on the test dataset with pseudo-labels, and measures the distribution discrepancy between the training and the test datasets in the parameter level:

$$S_{pro}(\mathcal{D}_{\text{test}}) = ||\tilde{\boldsymbol{\theta}}_{ref} - \tilde{\boldsymbol{\theta}}||_2,$$

where $\boldsymbol{\theta}_{ref}$ denotes the parameters of the pre-trained model, while $\boldsymbol{\theta}$ denotes the parameters of the fine-tuned model.

Those algorithms mentioned in this paper can be summarized as Table 5.

## D Formulation of the entropy loss for low-confidence samples

In the ablation study, we explore the impact of loss selection on the performance of unsupervised accuracy estimation. In particular, the detail about the entropy loss for samples with low confidence is expressed as follows: In particular, the entropy loss can be expressed as follows:

$$\mathcal{L}(f_{\boldsymbol{\theta}}(\tilde{\mathbf{x}})) = -\frac{1}{m_1} \sum_{i=1}^{m_1} \sum_{k=1}^{K} \tilde{y}_{i,con>\tau}^{(k)} \log s_{\boldsymbol{\omega}}^{(k)}(f_{\boldsymbol{g}}(\tilde{x}_i^{con>\tau}))$$
$$- \frac{1}{m_2} \sum_{i=1}^{m_2} \sum_{k=1}^{K} s_{\boldsymbol{\omega}}^{(k)}(f_{\boldsymbol{g}}(\tilde{\mathbf{x}}_i^{con\leq\tau})) \log s_{\boldsymbol{\omega}}^{(k)}(f_{\boldsymbol{g}}(\tilde{\mathbf{x}}_i^{con\leq\tau})),$$

where the first term denotes the cross-entropy loss calculated for samples with confidence larger than the threshold value $\tau$, the second term denotes the entropy loss for samples with lower confidence than $\tau$, and *con* means the sample confidence, $m_1$ and $m_2$ denote the total number of samples with higher confidence and lower confidence than $\tau$, respectively.

Table 5: Method property summary including whether this method belongs to self-training or training-free approaches, and if this method requires training data or specific model architectures.

| Method | Self-training | Training-free | Training-data-free | Architecture-requirement-free |
|---|---|---|---|---|
| Rotation | ✗ | ✓ | ✓ | ✗ |
| ConfScore | ✗ | ✓ | ✓ | ✓ |
| Entropy | ✗ | ✓ | ✓ | ✓ |
| Agreement | ✗ | ✓ | ✓ | ✗ |
| ATC | ✗ | ✓ | ✗ | ✓ |
| Fréchet | ✗ | ✓ | ✗ | ✓ |
| Dispersion | ✗ | ✓ | ✓ | ✓ |
| Nuclear | ✗ | ✓ | ✓ | ✓ |
| ProjNorm | ✓ | ✗ | ✓ | ✓ |
| Ours | ✓ | ✗ | ✓ | ✓ |

# E    Influence of the Calibration Error

Theoretically, the proposed pseudo-labeling strategy depends on how well the prediction probabilities are calibrated. However, in practice, we do not need to have a perfectly calibrated model as we employ a mixed strategy that assigns pseudo-labels to high-confidence examples and random labels to low-confidence ones (Sohn et al., 2020; Dong et al., 2021; Yu et al., 2022a).

To demonstrate it empirically, in Table 6, we provide the expected calibration error (ECE, Guo et al. (2017)) of ResNet18 depending on the difficulty of test data. For this, we test first on CIFAR-10 (ID), and then on CIFAR-10C corrupted by brightness across diverse severity from 1 to 5. We can see that ECE is very low for ID data and remains relatively low across all levels of corruption severity, which shows that ResNet is quite well-calibrated on CIFAR-10.

Table 6: Expected Error Calibration (ECE) of ResNet18 on CIFAR-10 (ID) and CIFAR-10C corrupted by brightness across diverse severity from 1 to 5.

| Corruption Severity | ID | 1 | 2 | 3 | 4 | 5 |
|---|---|---|---|---|---|---|
| ECE | 0.0067 | 0.0223 | 0.0230 | 0.0243 | 0.0255 | 0.0339 |

On the other hand, in the case of a more complex distribution shift like Office-31, we can see that the calibration error has increased noticeably (Table 7). It is interesting to analyze this result together with Figure 3(a) of the main paper, where we compared the results between the usual pseudo-labeling strategy and the proposed one. Although our method has room for improvement compared to the oracle method, it is also significantly better than "pseudo-labels", indicating that the proposed label generation strategy is less sensitive to the calibration error.

Table 7: Expected Error Calibration (ECE) of ResNet18 on Office-31 data set.

| Domain | DSLR (ID) | Amazon | Webcam |
|---|---|---|---|
| ECE | 0.2183 | 0.2167 | 0.4408 |

# F    Comparison to Nuclear Norm

Here, we compare our method to Nuclear Norm (Deng et al., 2023) across 3 types of distribution shifts with ResNet18, ResNet50, and WRN-50-. Results are shown in Table 8. From this table, we observe our method outperforms Nuclear Norm under synthetic shift and natural shift.

Table 8: Performance comparison on 11 benchmark datasets with ResNet18, ResNet50 and WRN-50-2, where $R^2$ refers to coefficients of determination, and $\rho$ refers to the absolute value of Spearman correlation coefficients (higher is better). The best results are highlighted in **bold**.

| Method | Network | CIFAR 100 | | TinyImageNet | | Office-Home | | Camelyon17 | | Entity-13 | | Entity-30 | |
|--------|---------|-----------|-----------|--------------|-----------|-------------|-----------|------------|-----------|-----------|-----------|-----------|-----------|
| | | $R^2$ | $\rho$ | $R^2$ | $\rho$ | $R^2$ | $\rho$ | $R^2$ | $\rho$ | $R^2$ | $\rho$ | $R^2$ | $\rho$ |
| Nuclear | ResNet18 | **0.989** | 0.995 | **0.983** | **0.994** | 0.692 | 0.783 | 0.858 | **1.000** | **0.978** | **0.991** | **0.980** | 0.993 |
| | ResNet50 | 0.979 | **0.994** | 0.965 | 0.994 | 0.731 | 0.895 | 0.849 | **1.000** | **0.989** | **0.996** | **0.978** | 0.994 |
| | WRN-50-2 | 0.962 | 0.988 | 0.956 | 0.992 | 0.766 | 0.874 | 0.983 | **1.000** | **0.989** | 0.995 | **0.985** | **0.996** |
| | Average | 0.977 | 0.993 | 0.968 | 0.993 | 0.730 | 0.850 | 0.916 | **1.000** | **0.989** | **0.995** | **0.989** | **0.995** |
| Ours | ResNet18 | 0.987 | **0.996** | 0.971 | **0.994** | **0.876** | **0.909** | **0.996** | 1.000 | 0.969 | **0.991** | 0.970 | **0.995** |
| | ResNet50 | **0.991** | **0.994** | **0.980** | **0.995** | **0.829** | **0.944** | **0.999** | 1.000 | 0.960 | 0.995 | 0.957 | **0.996** |
| | WRNt-50-2 | **0.995** | **0.998** | **0.975** | **0.996** | **0.809** | **0.916** | **0.997** | 1.000 | 0.968 | **0.995** | 0.949 | 0.994 |
| | Average | **0.991** | **0.997** | **0.976** | **0.995** | **0.837** | **0.923** | **0.998** | 1.000 | 0.966 | 0.994 | 0.959 | 0.995 |

Table 9: Performance comparison between GDSCORE from different layers on CIFAR-10C with ResNet18. The performance is measured by coefficients of determination (i.e., $R^2$).

| ResNet18 | Block 1 | Block 2 | Block 3 | Last layer | All |
|----------|---------|---------|---------|------------|-----|
| CIFAR-10 | 0.880 | 0.972 | 0.984 | 0.986 | 0.983 |

# G  Why are the gradients of the entire network not as predictive as that of the last layer?

We provide two intuitive explanations for this phenomenon, together with pointers to the existing body of work on our learning task, below:

- Deep neural networks detect general features at lower (e.g., first) layers and specific features (parts, objects) at higher (e.g., last) layers (Chen et al., 2021). Consequently, higher layers would be more sensitive to the changes of distribution shift than lower layers. So, it is reasonable that the last layer can capture more information about the distribution shift.

- The gradients are getting weaker and weaker as moving back through the hidden layers (Raghu et al., 2021). Therefore, the information on the distribution shift contained in the gradients may also decrease after backpropagation through layers.

This may explain why the gradients of the last layer can achieve the best for this task. Moreover, the gap between different layers can be affected by network architecture. For example, recent work (Raghu et al., 2021) shows that, compared to CNNs, ViTs have more uniform representations with greater similarity between lower and higher layers. The reason is that most information in ViT passes through skip connections. Therefore, it is natural that using different layers of ViT would not result in a large gap in performance in our method. This will not affect the effectiveness of the proposed method using the last layer.

To certify it, we conduct an ablation study of layer selection with ResNet18 and ViT in Table 9 and Table 10, respectively. Results shown below illustrate that the last layer performs satisfactorily for both ResNet models and transformer-based models. Furthermore, it is worth noting that the estimation performance of ViT is more consistent even with deeper layers compared to ResNet18.

# H  Proofs

## H.1  Proof of Theorem 4.1

We start by proving the following lemma.

Table 10: Performance comparison between GDSCORE from different layers on CIFAR-10C with ViT. The performance is measured by coefficients of determination (i.e., $R^2$).

| ViT | Block 1-18 | Block 9-16 | Block 17-24 | Last layer | All |
|---|---|---|---|---|---|
| CIFAR-10 | 0.913 | 0.953 | 0.935 | 0.948 | 0.938 |

**Lemma H.1.** *For any convex function $f : \mathbb{R}^D \to \mathbb{R}$ and any $p, q \geq 1$ such that $\frac{1}{p} + \frac{1}{q} = 1$, we have:*

$$\forall \mathbf{a}, \mathbf{b} \in dom(f), \quad |f(\mathbf{a}) - f(\mathbf{b})| \leq \max_{\mathbf{c} \in \{\mathbf{a}, \mathbf{b}\}} \{\|\nabla f(\mathbf{c})\|_p\} \cdot \|\mathbf{a} - \mathbf{b}\|_q.$$

*Proof.* Using the fact that $f$ is convex, we have:

$$\begin{aligned}
f(\mathbf{a}) - f(\mathbf{b}) &\leq \langle \nabla f(\mathbf{a}), \mathbf{a} - \mathbf{b} \rangle \\
&\leq |\langle \nabla f(\mathbf{a}), \mathbf{a} - \mathbf{b} \rangle| \\
&\leq \sum_{i=1}^{p} |\nabla f(\mathbf{a})_i (\mathbf{a}_i - \mathbf{b}_i)| \\
&\leq \|\nabla f(\mathbf{a})\|_p \|\mathbf{a} - \mathbf{b}\|_q,
\end{aligned}$$

where we used Hölder's inequality for the last inequality. The same argument gives:

$$f(\mathbf{b}) - f(\mathbf{a}) \leq \|\nabla f(\mathbf{b})\|_p \|\mathbf{b} - \mathbf{a}\|_q.$$

Using the absolute value, we can combine the two previous results and obtain the desired inequality. □

The proof of Theorem 4.1 follows from applying Lemma H.1 to the convex function $\mathcal{L}_T$.

## H.2 Proof of Theorem 4.2

*Proof.* The proof follows from Theorem 4.1 by noting that $\|\boldsymbol{\omega}_s - \mathbf{c}\|_q = \eta \|\nabla \mathcal{L}_T(\boldsymbol{\omega}_s)\|_q$. □

## H.3 Proof of Theorem 4.3

We start by introducing some notations. We denote $\mathcal{L}_{\mathbf{x},y}$ the loss evaluated on a specific data-point $(\mathbf{x}, y) \sim P_T(\mathbf{x}, \mathbf{y})$. We can then decompose the expected loss as $\mathcal{L}_T = \mathbb{E}_{P_T(\mathbf{x},y)} \mathcal{L}_{\mathbf{x},y}$. It follows by linearity of the expectation that

$$\nabla \mathcal{L}_T = \mathbb{E}_{P_T(\mathbf{x},y)} \nabla \mathcal{L}_{\mathbf{x},y}.$$

Then, we prove the following lemma that gives the formulation of the gradient of the cross-entropy.

**Lemma H.2.** *The gradient of the cross-entropy loss with respect to $\boldsymbol{\omega} = (\mathbf{w}_k)_{k=1}^K$ writes*

$$\nabla \mathcal{L}_{\mathbf{x},y}(\boldsymbol{\omega}) = \left( -y^{(k)} \mathbf{x}(1 - s_{\boldsymbol{\omega}}^{(k)}(\mathbf{x})) \right)_{k=1}^K.$$

*Proof.* First, let's compute the partial derivative of the softmax w.r.t. $\mathbf{w}_k$ for any $k \in \{1, \dots, K\}$. We have:

$$\begin{aligned}
\frac{\partial s_{\boldsymbol{\omega}}^{(k)}(\mathbf{x})}{\partial \mathbf{w}_k} &= \frac{\mathbf{x} e^{\mathbf{w}_k^\top \mathbf{x}} \left( \sum_{\tilde{k}} e^{\mathbf{w}_{\tilde{k}}^\top \mathbf{x}} - e^{\mathbf{w}_k^\top \mathbf{x}} \right)}{\left( \sum_{\tilde{k}} e^{\mathbf{w}_{\tilde{k}}^\top \mathbf{x}} \right)^2} \\
&= \mathbf{x} \left( s_{\boldsymbol{\omega}}^{(k)}(\mathbf{x}) - \left[ s_{\boldsymbol{\omega}}^{(k)}(\mathbf{x}) \right]^2 \right).
\end{aligned}$$

Using the chain rule, the partial derivative of the loss w.r.t. $\mathbf{w}_k$ writes:

$$\frac{\partial \mathcal{L}_{\mathbf{x},y}(\boldsymbol{\omega})}{\partial \mathbf{w}_k} = \frac{\partial \mathcal{L}_{\mathbf{x},y}(\boldsymbol{\omega})}{\partial s(\boldsymbol{\omega},\mathbf{x})} \cdot \frac{\partial s(\boldsymbol{\omega},\mathbf{x})}{\partial \mathbf{w}_k}$$

$$= \begin{cases} -\frac{1}{s_{\boldsymbol{\omega}}^{(k)}(\mathbf{x})} \cdot \mathbf{x} \left( s_{\boldsymbol{\omega}}^{(k)}(\mathbf{x}) - \left[ s_{\boldsymbol{\omega}}^{(k)}(\mathbf{x}) \right]^2 \right), & \text{if } y^{(k)} = 1 \\ 0, & \text{otherwise} \end{cases}$$

$$= -y^{(k)}\mathbf{x}\left(1 - s_{\boldsymbol{\omega}}^{(k)}(\mathbf{x})\right)$$

As the $\frac{\partial \mathcal{L}_{\mathbf{x},y}(\boldsymbol{\omega})}{\partial \mathbf{w}_k}$ are the coordinates of $\nabla \mathcal{L}_{\mathbf{x},y}(\boldsymbol{\omega})$, we obtain the desired formulation. $\square$

We now proceed to the proof of Theorem 4.3.

*Proof.* Using the convexity of $\|\cdot\|_p$ and the Jensen inequality, we have that

$$\|\nabla \mathcal{L}_T(\boldsymbol{\omega})\|_p = \|\mathbb{E}_{P_T(\mathbf{x},y)} \nabla \mathcal{L}_{\mathbf{x},y}(\boldsymbol{\omega})\|_p$$

$$\leq \mathbb{E}_{P_T(\mathbf{x},y)} \|\mathcal{L}_{\mathbf{x},y}(\boldsymbol{\omega})\|_p \qquad \text{(Jensen inequality)}$$

$$= \mathbb{E}_{P_T(\mathbf{x},y)} \left( \sum_{i=1}^{D} \sum_{k=1}^{K} |-y^{(k)}\mathbf{x}_i(1 - s_{\boldsymbol{\omega}}^{(k)}(\mathbf{x})|^p \right)^{1/p}$$

$$= \mathbb{E}_{P_T(\mathbf{x},y)} \left( \sum_{k=1}^{K} y^{(k)} \left(1 - s_{\boldsymbol{\omega}}^{(k)}(\mathbf{x})\right)^p \right)^{1/p} \left( \sum_{i=1}^{D} |\mathbf{x}_i^p| \right)^{1/p}$$

$$= \mathbb{E}_{P_T(\mathbf{x},y)} \left( \left(1 - s_{\boldsymbol{\omega}}^{(k_y)}(\mathbf{x})\right)^p \right)^{1/p} \left( \sum_{i=1}^{D} |\mathbf{x}_i^p| \right)^{1/p} \qquad (k_y \text{ such that } y^{(k_y)} = 1)$$

$$= \mathbb{E}_{P_T(\mathbf{x},y)} \alpha(\boldsymbol{\omega},\mathbf{x},y)\|\mathbf{x}\|_p,$$

where $\alpha(\boldsymbol{\omega},\mathbf{x},y) = \left(1 - s_{\boldsymbol{\omega}}^{(k_y)}(\mathbf{x})\right)$, with $k_y$ such that $y^{(k_y)} = 1$. We used the fact that $\mathbf{y}$ is a one-hot vector so it has only one nonzero entry. $\square$

## H.4 Proof of Remark 6.1

*Proof.* Using the reverse Minkowski inequality, as $0 < p < 1$, we have that

$$\|\boldsymbol{\omega}_s\|_p = \|\mathbf{c} + \eta \cdot \nabla \mathcal{L}_T(\boldsymbol{\omega}_s)\|_p \geq \|\mathbf{c}\|_p + \eta \cdot \|\nabla \mathcal{L}_T(\boldsymbol{\omega}_s)\|_p$$
$$\implies \|\boldsymbol{\omega}_s\|_p - \|\mathbf{c}\|_p \geq \eta \cdot \|\nabla \mathcal{L}_T(\boldsymbol{\omega}_s)\|_p.$$

In the same fashion, we have that

$$\|\mathbf{c}\|_p = \|\boldsymbol{\omega}_s - \eta \cdot \nabla \mathcal{L}_T(\boldsymbol{\omega}_s)\|_p \geq \|\boldsymbol{\omega}_s\|_p + \eta \cdot \|\nabla \mathcal{L}_T(\boldsymbol{\omega}_s)\|_p$$
$$\implies \|\mathbf{c}\|_p - \|\boldsymbol{\omega}_s\|_p \geq \eta \cdot \|\nabla \mathcal{L}_T(\boldsymbol{\omega}_s)\|_p.$$

We obtain the desired upper bound by combining those results. $\square$

