# OpenReview forum: "Leveraging Gradients for Unsupervised Accuracy Estimation under Distribution Shift"
_TMLR — Accepted by TMLR_

### Review · Reviewer_fmXC · 2024-11-05

**Summary Of Contributions:**

In the context of assessing the performance of multi-class classification neural networks without access to ground-truth labels and under potential distribution shifts, the paper introduces GdScore, a novel statistic based on the norm of the gradients from the classification layer after one backpropagation step on test data. The authors show that this norm strongly correlates with test accuracy, offering a new and more efficient alternative to traditional methods.

The process works as follow:
- Generate labels for the test data with a pseudo-labeling method based of prediction confidence
- Using these labels, compute the gradient of the weights of the classification layer
- Take the Lp norm of this gradient

After a theoretical analysis and a formal introduction of the GdScore method, the paper presents empirical results on benchmark datasets with various types of distribution shifts in the test data. The testing shows that GdScore correlates well with test accuracy and outperforms the benchmark metrics.

**Audience:**

Yes

**Claims And Evidence:**

Yes

**Requested Changes:**

See weaknesses section above. In short:
- Further discuss the implications of the modeling assumptions made as well as their limitations / tradeoffs
- Add further insights for alternative performance measures to get a more comprehensive view of how GdScore can help assess generalization
- Paper and its organization can be confusing at times - e.g. related work at the complete end of the paper
- Unclear what the ablation study is exactly - was not introduced previously
- Add code for reproducibility

**Strengths And Weaknesses:**

Strengths:
- GdScore approach is elegant and intuitive, and results look very promising as well as more efficient
- The self-labeling mechanism is an interesting method and can be reused to other problems
- Paper is rigorous and all claims are justified theoretically and empirically
- The paper provides meaningful qualitative and quantitative insights on the behavior of GdScore and other methods, as well as their sensitivity to different parameter choices

Weaknesses:
-  Assumption that classification errors mainly comes from the low confidence predictions is very strong, particularly in a distribution shift context. While it might hold for baseline datasets, I feel the paper does not discuss enough this assumption as well as the limitation it brings.
- I would have wanted to see other measures of classification performance that would help assess the generalization capabilities of a given network, e.g. precision and recall in a binary classification case
- The paper does not discuss enough the limitations of the metric presented - and therefore lacks nuance

---

> ### Author Response · Authors · 2024-11-21
>
> We thank the reviewer fmXC for the constructive suggestions and positive support of this work. We appreciate that the reviewer found our work interesting and theoretically and empirically strong.
>
> We address the reviewer's concerns point by point below.
>
> >**Q1: Further discuss the implications of the modeling assumption that classification errors mainly come from the low-confidence predictions.**
>
> We thank the reviewer for suggesting further discussion about our label-generation strategy. We provide additional discussion below that will be added to the revised version of the paper.
>
> Initially, it is important to note that we do not consider samples with low confidence as **definitively** misclassified samples. Rather, we view them as **more potentially** misclassified. Some empirical phenomena also support this assumption, where misclassified samples are significantly closed to decision boundaries, therefore with lower confidence, than the correctly classified ones [1]. This is why we allocate random labels to low-confidence samples.
>
> * *[The implication of designating a greater or lesser number of samples as misclassified.]*  In Table 2, we illustrate the implication of assigning more or less samples as misclassified ones via adjusting the value of $\eta$. From this table, we can see that if we treat all pseudo labels as the correct (i.e., $\eta$=0), the performance will decrease compared with our labeling strategy. A similar situation also happens if we treat all pseudo labels as misclassified (i.e., $\eta$=1). It implies that we should carefully choose the threshold of low confidence.
> * *[Why do we choose $\eta=0.5$?]* In this work, we set the value of $\tau$ as 0.5 across all datasets and network architectures due to an intuitive reason. If a label contains a softmax probability below 0.5, it means that this predicted label has over 50% probability of being wrong. It means that this label has a higher probability of being incorrect than to be correct. Thus, we tend to regard it as an incorrect prediction.
>
> * *[The limitation of our label generation strategy.]* We agree with the reviewer that misclassified samples are not always with low confidence. Empirical evidence in Figure 3 also reinforces this point. The figure demonstrates that performance is superior when utilizing true labels compared to our label generation strategy, despite the fact that our strategy's performance ranks second. Therefore, the limitation of our proposed strategy is that it cannot discover misclassified samples with high confidence as well as correct samples with low confidence. However, our approach is notably efficient and requires less computational resources when contrasted with the alternative labeling strategies depicted in Figure 5.
>
> > **Q2: Add further insights for alternative performance measures such as precision and recall.**
>
> We thank the reviewer for their suggestion. We would first like to clarify that the goal of this field is to estimate ground-truth accuracy on test data with distribution shifts in an unsupervised manner. As other performance measures and test accuracy are not a one-to-one correspondence, we mainly focus on the absolute value of Spearman correlation coefficients (i.e., $\rho$) between GdScore and the performance measurements. The numerical results are shown as follows.
>
> **Table A: $\rho$ on CIFAR-10C between GdScore and diverse measure metrics.**
> |Metrics|Conf|Entropy|ATC|Frechet|Dispersion|GdScore (Ours)|
> | --- | - | - | -|-|-|-|
> |Precision|0.902|0.904|0.902|0.860|0.898|**0.958**|
> |Recall|0.842|0.848|0.859|0.825|0.881|**0.959**|
> |F1 score|0.964|0.966|0.974|0.939|**0.988**|**0.985**|
>
> In Table A, we regard the first class as the positive class while the other as the negative. From this table, we observe that GdScore also performs well in estimating the other measure metrics.
>
> > **Q3: Discuss about the limitations of the GdScore.**
>
> We discuss below the limitations of the GdScore and will add this paragraph in the revised version of the paper.
>
> Although GdScore is a source-free and one-step backpropagation approach without requirements for specific model architectures, it also has certain limitations which should be addressed in future work.
> * *[Computational cost should be reduced.]* We note that our approach is more computationally efficient than multi-step self-training methods such as ProjNorm [2], and methods requiring large-scale training data like Frechet [3] and ATC [4]. However, although our method is more stable and performs better, the computational efficiency of GdScore still needs to be improved compared with training-free and source-free methods such as Entropy [5] and Nuclear [6].

---

> ### Author Response · Authors · 2024-11-21
>
> > **Q4: Explanation about the ablation study.**
>
> We apologize for the lack of clarity in our ablation study. In general, we explore the impact of hyperparameters (i.e., gradient norm type $p$ and threshold in our label generation strategy $\eta$), the self-training process including epochs and losses, the layer to calculate gradients and calibration scenarios on the final performance. We will refine this part in our final version so that it can be easily understood.
>
> > **Q5: About the organization and the code.**
>
> We acknowledge the reviewer's comment and we will refine the organization in the revised version of the paper. Regarding the code, we upload it using an anonymous Google Drive which link is: https://drive.google.com/file/d/1wSjIqoe5Ibkx_qAEFqHoJp3tkSyAtvog/view?usp=sharing and we will open-source it once it's accepted.
>
> We hope that all the reviewer's concerns and questions have been addressed and we remain open to future discussions.
>
> [1] Mickisch, D., Assion, F., Greßner, F., Gunther, W., and Motta, M. (2020). Understanding the decision boundary of deep neural networks: An empirical study. arXiv preprint arXiv:2002.01810.
>
> [2] Yaodong Yu, Zitong Yang, Alexander Wei, Yi Ma, and Jacob Steinhardt. Predicting out-of-distribution error with the projection norm. arXiv preprint arXiv:2202.05834, 2022b.
>
> [3] Weijian Deng and Liang Zheng. Are labels always necessary for classifier accuracy evaluation? In Proceedings of the IEEE/CVF Conference on Computer Vision and Pattern Recognition (CVPR), pp. 15069–15078, 2021.
>
> [4] Saurabh Garg, Sivaraman Balakrishnan, Zachary C Lipton, Behnam Neyshabur, and Hanie Sedghi. Leveraging unlabeled data to predict out-of-distribution performance. arXiv preprint arXiv:2201.04234, 2022.
>
> [5]Devin Guillory, Vaishaal Shankar, Sayna Ebrahimi, Trevor Darrell, and Ludwig Schmidt. Predicting with confidence on unseen distributions. In Proceedings of the IEEE/CVF International Conference on Computer Vision (ICCV), pp. 1134–1144, 2021
>
> [6]Deng W, Suh Y, Gould S, et al. Confidence and dispersity speak: Characterizing prediction matrix for unsupervised accuracy estimation[C]//International Conference on Machine Learning. PMLR, 2023: 7658-7674.

---

### Review · Reviewer_LHuS · 2024-11-18

**Summary Of Contributions:**

This work studies how gradients can be leveraged for accuracy estimation under the assumption of distribution shift. More specifically, the authors propose to use the norm of the classification-layer gradients and backpropaged from the cross-entropy loss for only one step, and use that information to estimate a score which reveals the confidence of the model in the new test dataset under the unknown distribution shift.

**Audience:**

Yes

**Broader Impact Concerns:**

No broader impact concerns

**Claims And Evidence:**

Yes

**Requested Changes:**

See the weakness part of the work in 'Strengths and weaknesses' section

**Strengths And Weaknesses:**

There are multiple strengths of this work.

First of all, the draft works on both theoretical and empirical justification, which makes the draft look stronger than a pure theoretical work or pure empirical work.

Second, unlike existing works that mostly require the ground-truth label information in the test set to understand if the model is robust for the desired test sets, the proposed methods will not be constrained by lacking ground truth labels.

Finally, The presented method, called GdScore, does show very high correlation with the test accuracy, and is clearly stronger than the baselines that it is compared to.





Weakness :

This work uses the gradient from the classification layer, which imposes clear limitations to this work. For example, can this work be applied to foundation models which are not only favoring classification tasks? Can this work be favoring the models that are beyond the classification tasks?

In the continual learning domain, there are a lot of works that achieve the continual learning via adapting the layer normalization layers. That been said, two directions the authors can consider:

-- For a large model where layer normalization, batch normalization are applied layer to layer, will the proposed methods still be significant? Assuming the model will be trained to do classification.

-- Can the proposed methods be adapted to handle the normalization layers of the transformer?

---

> ### Author Response · Authors · 2024-11-21
>
> We thank the reviewer for their positive support and the precious comments to help us improve our paper. We hope our following responses will address the reviewer's concerns precisely.
>
> > **Q1: Can this work be applied to foundation models that are not only favoring classification tasks? Can this work favor the models that are beyond the classification tasks?**
>
> We thank the reviewer for their suggestion. We first note that the main focus of our work is to estimate the test accuracy of a model under distribution shifts. This is the reason why we focused on classification tasks that encompass a broad range of applications. That being said, from a methodological point of view, our method relies on the information contained in gradients based on loss minimization. Hence, it could be interesting for future work to apply that on LLMs for instance where the task is next token prediction, or on other foundations models. Intuitively, this should work as gradients still contain information about the classification task, although some other part of the method (e.g., the pseudo-labeling) might have to be adapted.
>
>
> > **Q2: Will the proposed methods still be significant in large models?**
>
> We appreciate the reviewer for introducing this interesting question. This question depends on the architecture of the large model. For example, GdScore is hard to adapt in Clip due to the lack of a classification layer. If the model architecture includes the final classification layer, from a methodological point of view, having access to the gradients of the last layer can still be done in such cases. Experiments and ablations would have to be conducted to determine whether the normalization layers should be taken into account for the gradient computations.
>
> > **Q3: Can the proposed methods be adapted to handle the normalization layers of the transformer??**
>
> We thank the reviewer for the suggestion. In the table below, we illustrate $R^2$ between GdScore calculated from different blocks and the test accuracy.
> |Layer|Block 1|Block 2|Block 3|Last layer|All|
> | --- | - | - | - | - | - |
> |TinyImageNet|0.625|0.944|0.938|**0.971**|0.949|
> |Office-31|**0.749**|0.507|0.453|0.675|0.688|
> |Office-Home|0.486|0.569|0.499|**0.875**|0.652|
> |Fmov|0.269|0.238|0.245|**0.687**|0.272|
>
> From this table, we can see that the gradients from the last classification layer estimate the most precise test accuracy. Therefore, we believe that our approach cannot easily be adapted to the normalization layer of Transformer. However, our method can be used straightforwardly to estimate the test accuracy of transformer-based models. The results are shown below.
> |ConvNeXt|Dispersion|Nuclear|COT|Ours|
> | --- | - | - | -|-|
> |Cifar10.1|11.312|4.667|2.236|**0.967**|
> |ImageNet-S|5.751|2.493|4.084|**1.033**|
>
> The results are conducted with ConvNeXt for classification, where we only compute the gradient w.r.t the classification layer.
>
> We thank the reviewer for their questions that provide us with very interesting and important future work. We hope that all the reviewer's concerns and questions have been addressed and we remain open to future discussions.

---

### Review · Reviewer_R4ZR · 2025-02-04

**Summary Of Contributions:**

This paper proposes a method for unsupervised performance estimation under distribution shift. The authors show how gradient information can be a good predictive measure of test accuracy under such shifts. To avoid using ground truth test labels, they use a pseudo-labelling technique that basically uses the predicted label if the model is confident or a random one otherwise. The authors motivate their proposed gradient based measure with some simple theory. The evaluation is performed over a bunch of different distribution shifts for different pre-training datasets, and the authors compare their results to several baselines highlighting the goodness of their metric.

**Audience:**

Yes

**Claims And Evidence:**

Yes

**Requested Changes:**

Please refer to the questions in the section above.

**Strengths And Weaknesses:**

The paper is generally easy to read and follow. The evaluation of the metric is very comprehensive, and shows that this simple measure is indeed a pretty decent proxy for test accuracy under certain kinds of distribution shifts.

I have a few questions that I will be listing here.

**Clarifying question re the approach**: The method just looks at the gradient of the outer layer of the pre-trained model using the soft-labelling described in sec. 4.1, and there is no update of weights using this gradient, right? I am asking this because throughout the intro and methodology section, that’s what the writing looks like. However, on page 9 “GdScore is a more efficient self-training approach” paragraph, the text says “our method only trains the model for one epoch….” If yes (assuming no updates are made), in Corollary 4.2 shouldn’t my main quantity of concern be $L_T(w_s)$ bc we want to get a proxy for $0/1$ error of the pre-trained model $w_s$? What is this telling me, then? If I were to take a step in that direction, the difference in loss would be small if my gradient norm at $w_s$ is controlled? Essentially saying that the model at $w_s$ was already good enough? Please clarify this a little.



**Re the pseudo labelling**: In a very generic sense, you assume that if the source and train distribution are close enough, you’d probably get good pseudo-labels and random ones if they’re far apart (assuming the model’s confidence terrible, and uniform-ish). However, I could make adversarial distributions, for which the model is very confident with terrible predictions. Any comments on these kinds of distribution shifts?

**Comparisons to other methods**: For all these methods, are the models they use also pre-trained on Imagenet, and then trained until same (similar) in distribution accuracy as your method? This feels like a very crucial detail that is missing.


**Figure 5b)**: It is kinda odd that gradients of the entire network are not as predictive as that of the last layer. Why do you think that is the case?

**Figure 5c)** is kind of an obvious result you’d expect (I think it also clarifies my Question 1? You’re just storing the gradients after 1 epoch, but please clarify). Any fine-tuning on the model won’t be predictive of the pre-trained model’s performance. But it is kind of surprising to see how fast, for CIFAR-100 for example, this predictive nature vanishes. Also, kind of surprising that it saturates for the other two datasets in the image and doesn't degrade any further. Some comment and discussion on this would be helpful.

Minor: Please mention the GPUs you used, the compute time, all these details. Also, are the results averaged across different seeds? Please mention that too.

---

> ### Author Response · Authors · 2025-02-09
>
> We thank the reviewer for their positive comments, identifying potential confusing points, and providing us with constructive suggestions. We hope the following responses can effectively address their concerns.
>
> >**Q1: Clarifying questions about the approach.**
>
> The reviewer is indeed right: there is no update of weights using the gradients. We thank the reviewer for pointing out this lack of clarity and will update the manuscript accordingly.
>
> The point of Corollary 4.2 is to show how the pseudo-labeling connects to the gradient norm used for the pexory for error of the pre-trained model. More precisely, it shows how the gradient norm is informative of the quality os pseudo-labels and hence of the performance of the pre-trained model on the test data (indeed, if the pre-trained model pseudo-labels perfectly, its accuracy on target data would be very high and vice-versa). The reviewer is correct: taking a step in the direction of c, the loss difference would be small if the gradient norm is controlled. This amounts to say that the pre-trained model (that has classifier $w_s$) pseudo-labels the target samples well, which implies its test accuracy is high. In summary, a high gradient would indicate a poor pseudo-labelling and hence a small test accuracy and vice-versa, which is confirmed in our experiments.
>
> Going back to the reviewer's first point, we use the terms "backpropagation" to say that if the model was to be updated, it would implies a high or small gradient update, which in turns thanks to Corollary 4.2 is connected to the loss update, but we do not update the pre-trained model in practice. We will clarify this point in the updated manuscript.
>
> >**Q2: Re the pseudo labelling**
>
> We thank the reviewer for reminding us of the adversarial setting. In general, we agree with the reviewer that adversarial distributions can be reviewed as the worst case of all distribution shifts; however, there are differences between adversarial distributions and the distribution shifts we discussed. We clarify our approach under the adversarial setting in two respects.
>
> * *[Adversarial distributions are different from traditional distribution shifts derived from domain generalization [1].]* In the field of domain generalization, we mainly consider three types of distribution shifts: the covariate shift (i.e., $P_{train}(x)\neq P_{test}(x), P_{train}(y|x)=P_{test}(y|x)$), label shift (i.e., $P_{train}(y)\neq P_{test}(y), P_{train}(x|y)=P_{test}(x|y)$), and concept shift (i.e., $P_{train}(y|x)\neq P_{train}(y|x), P_{train}(x)=P_{test}(x)$), While adversarial distributions are $P_{train}(x)\neq P_{test}(x), P_{train}(y|x) \neq P_{test}(y|x)$. Under the adversarial setting, this difference can lead to the degeneration of approaches designed for domain-generalization distribution shifts due to different data corruption mechanisms [2]. Similarly, $\texttt{GdScore}$ is also specially designed for domain-generalization distribution shifts.
>
> * *[Assign trustworth labels as psuedo labels based on low-density assumption.]*
> We assign pseudo labels based on the confidence of the model due to the low-density assumption usually used in self-training and semi-supervised learning [3], which states that optimal decision boundaries should lie in low-density regions. The assumption is often empirically supported as the misclassified samples tend to be significantly closer to the decision boundary than the correctly classified ones [4], which implies that correctly classified samples are prone to have higher confidence than wrongly classified ones. However, we believe that this assumption is invalid under the adversarial setting since adversarial perturbations assign instances to the wrong groups in the hidden feature space and enlarge the distance among those groups with the increase of perturbation size [2].
>
> >**Q3: Comparisons to other methods**
>
> Yes, all the baselines and our approach use the same pre-trained models across different distribution shifts and model architectures. In particular, the pre-trained models are trained on the CIFAR-10, CIFAR-100, ImageNet, TinyImageNet, each domain in Office31 and Office-Home, the training set of Camelyon17-WILDS, and BREEDS conducted from ImageNet. For all methods, the model's in-distribution accuracy is the same since it does not depend on the method (it is the estimation score that changes between methods). This is the usual experimental setup adopted in prior works. We apologize for the lack of clarity and will add this explanation to the updated manuscript.

---

> ### Author Response · Authors · 2025-02-09
>
> **Q4: It is kinda odd that gradients of the entire network are not as predictive as that of the last layer. Why do you think that is the case?**
>
> We provide two intuitive explanations, together with pointers to the existing body of work on our learning task, below:
>
> * Deep neural networks detect general features at lower (e.g., first) layers and specific features (parts, objects) at higher (e.g., last) layers [5]. Consequently, higher layers would be more sensitive to the changes of distribution shift than lower layers. So, it is reasonable that the last layer can capture more information about the distribution shift.
>
> * The gradients are getting weaker and weaker as moving back through the hidden layers [6]. Therefore, the information on the distribution shift contained in the gradients may also decrease after backpropagation through layers.
>
> This may explain why the gradients of the last layer can achieve the best for this task. Moreover, the gap between different layers can be affected by network architecture. For example, recent work [6] shows that, compared to CNNs, ViTs have more uniform representations with greater similarity between lower and higher layers. The reason is that most information in ViT passes through skip connections. Therefore, it is natural that using different layers of ViT would not result in a large gap in performance in our method. This will not affect the effectiveness of the proposed method using the last layer.
>
> To certify it, we conduct an ablation study of layer selection with ResNet18 and ViT, respectively. Results shown below illustrate that the last layer performs satisfactorily for both ResNet models and transformer-based models. Furthermore, it is worth noting that the estimation performance of ViT is more consistent even with deeper layers compared to ResNet18.
>
> |CIFAR-10|Block 1|Block 2|Block 3|Last layer|All|
> | --- | - | - | - | - | - |
> |ResNet18|0.880|0.972|0.984|**0.986**|0.983|
>
> |CIFAR-10|Block 1-8|Block 9-16|Block 17-24|Last layer|All|
> | --- | - | - | - | - | - |
> |ViT|0.913|0.953|0.935|0.948|0.938|
>
>
> **Q5: Some comment and discussion on Figure 5c.**
>
> The reviewer is right: we indeed just store the gradients after 1 epoch (see answer to Question 1 above). We will add this clarification in the updated manuscript.
>
> Regarding the decrease in predictive power, gradients in the first epoch contain the most abundant information about the training dataset, which will be deluded along the further epochs (a model updates a lot on the test data would forget previous training categories along the iterations). Regarding the difference of decrease between the image datasets, one could think that seeing too much test data would delude faster the training information for CIFAR-100 since it has less training example per class than CIFAR-10 and more than TinYImageNet but with lower resolution (32x32 vs 64x64). Once the "training" information of the classifier is somewhat lost, it does not decrease more since the pre-trained model still contains valuable training information in its feature extractor. We will add this discussion to the updated manuscript.
>
> **Q6: Computational resources**
>
> We conduct all the experiments on NVIDIA 4090. Since no training is involved (no randomization of the batches for the stochasticity of SGD) and test data are not subsampled, we use the same experimental setting as prior works and always fix the seed as 1 for all baselines and our approach during evaluation to ensure fair comparison. We also list the partial compute time (i.e., $T$, which refers to average evaluation time. Its unit is seconds) results below. We thank the reviewer for their feedback and will add this information in the updated manuscript.
>
> |$T$|CIFAR-10|CIFAR-100|Office-31|Office-Home|Entity30|Living17|
> | --- | - | - | - | - | - |  - |
> |Ours|47.207|47.174|4.475|14.559|32.882|32.660|
>
> [1]Alhamoud K, Hammoud H A A K, Alfarra M, et al. Generalizability of adversarial robustness under distribution shifts[J]. arXiv preprint arXiv:2209.15042, 2022.
>
> [2]Xie R, Wei H, Feng L, et al. On the importance of feature separability in predicting out-of-distribution error[J]. Advances in Neural Information Processing Systems, 2024, 36.
>
> [3]Chapelle, O. and Zien, A. (2005). Semi-supervised classification by low density separation. In Proceedings of the Tenth International Workshop on Artificial Intelligence and Statistics, pages 57–64.
>
> [4]Mickisch, D., Assion, F., Greßner, F., Gu ̈nther, W., and Motta, M. (2020). Understanding the decision boundary of deep neural networks: An empirical study. arXiv preprint arXiv:2002.01810.
>
> [5] Chen, Jiefeng, et al. "Detecting errors and estimating accuracy on unlabeled data with self-training ensembles." Advances in Neural Information Processing Systems 34 (2021): 14980-14992.
>
> [6] Raghu, Maithra, et al. "Do vision transformers see like convolutional neural networks?." NeurIPS 2021.

---

> > ### Comment · Reviewer_R4ZR · 2025-02-15
> > **Response to authors**
> >
> > Thank you to the authors for their detailed response. I've read through all of them, I'm happy with all the clarifications, please add all the necessary ones to the updated manuscript. Also, the ViT vs. ResNet ablation is very nice, thanks for doing that. It would be great if you could add to the new version as well (the appendix is fine).

---

> > > ### Author Response · Authors · 2025-02-18
> > >
> > > Thank you for your valuable suggestions! We have incorporated them into the appendix of our latest version. We appreciate your continued positive support for our paper!

---

### Author Response · Authors · 2025-04-11
**Timeline for further discussion or recommendation**

Dear Action Editor,

Thank you for coordinating the review of our submission. We have received all three reviews, which expressed overall positive opinions and provided constructive suggestions to improve the paper. We have submitted detailed responses to each reviewer; since then, only one has acknowledged our rebuttal, while the other two, unfortunately, have not responded.

According to TMLR guidelines, the discussion period ended on March 4th, so we were wondering if there are any updates on the timeline for further discussion or a final recommendation.

We would be happy to provide any additional clarifications if needed, and we are grateful to you and the reviewers for the time taken and the valuable feedback.

Best regards,

The Authors

---

### Decision · Action_Editor_CHWx · 2025-04-12

**Recommendation:** Accept as is

**Comment:**

All reviewers found the paper interesting and well written. The paper is complete as it provides both a theory and complementary, well conducted, experiments. The authors-reviewers discussion have been fruitful and the modifications asked by the reviewers were implemented and seem to have further raised the quality of the manuscript.
Given all this, I recommend publication as is.

**Audience:**

I also agree with the reviewers that the paper addresses problems of interests to part of the TML audience.

**Claims And Evidence:**

All reviewers agree that the papers provide enough claims and evidence to meet the TMLR standards.